# High-dose vitamin D versus placebo to prevent complications in COVID-19 patients: Multicentre randomized controlled clinical trial

Javier Mariani[1,2]*, Laura Antonietti[1,2], Carlos Tajer[1,2], León Ferder[3], Felipe Inserra[3], Milagro Sanchez Cunto[4], Diego Brosio[5], Fernando Ross[6], Marcelo Zylberman[7], Daniel Emilio López[8], Cecilia Luna Hisano[9], Sebastián Maristany Batisda[1], Gabriela Pace[10], Adrián Salvatore[11], Jimena Fernanda Hogrefe[12], Marcela Turela[13], Andrés Gaido[14], Beatriz Rodera[15], Elizabeth Banega[16], María Eugenia Iglesias[17], Mariela Rzepeski[18], Juan Manuel Gomez Portillo[19], Magalí Bertelli[4], Andrés Vilela[6], Leandro Heffner[7], Verónica Laura Annetta[5], Lucila Moracho[4], Maximiliano Carmona[11], Graciela Melito[3], María José Martínez[1], Gloria Luna[1], Natalia Vensentini[1], Walter Manucha[20]

1 Hospital de Alta Complejidad en Red El Cruce—Néstor Kirchner, Florencio Varela, Buenos Aires, Argentina, 2 Universidad Nacional Arturo Jauretche, Florencio Varela, Buenos Aires, Argentina, 3 Maimónides University, Ciudad Autónoma de Buenos Aires, Buenos Aires, Argentina, 4 Hospital de Infecciosas Francisco Javier Muñiz, Ciudad Autónoma de Buenos Aires, Buenos Aires, Argentina, 5 Hospital General de Agudos Dr. Enrique Tornú, Ciudad Autónoma de Buenos Aires, Buenos Aires, Argentina, 6 Clínica Santa Isabel, Ciudad Autónoma de Buenos Aires, Buenos Aires, Argentina, 7 Hospital General de Agudos Dr. Cosme Argerich, Ciudad Autónoma de Buenos Aires, Argentina, 8 Hospital General de Agudos Dr. Teodoro Álvarez, Ciudad Autónoma de Buenos Aires, Buenos Aires, Argentina, 9 Hospital General de Agudos Dr. Juan A. Fernández, Ciudad Autónoma de Buenos Aires, Buenos Aires, Argentina, 10 Hospital General de Agudos Parmenio Piñero, Ciudad Autónoma de Buenos Aires, Buenos Aires, Argentina, 11 Hospital Luis Lagomaggiore, Mendoza, Argentina, 12 Sanatorio Güemes, Ciudad Autónoma de Buenos Aires, Buenos Aires, Argentina, 13 Hospital Regional Antonio J. Scaravelli, Tunuyán, Mendoza, Argentina, 14 Sanatorio Allende, Sede Cerro, Córdoba, Argentina, 15 Hospital Zonal General de Agudos "Dr. Isidoro Iriarte", Quilmes, Buenos Aires, Argentina, 16 Hospital Interzonal Especializado en Agudos y Crónicos, Neuropsiquiátrico Dr. Alejandro Korn, Melchor Romero, Buenos Aires, Argentina, 17 Sanatorio Allende Nueva Córdoba, Córdoba, Argentina, 18 Hospital Modular de Florencio Varela, Florencio Varela, Buenos Aires, Argentina, 19 Hospital El Carmen, Godoy Cruz, Mendoza, Argentina, 20 Consejo Nacional de Investigaciones Científicas y Técnicas, Universidad Nacional de Cuyo, Instituto de Medicina y Biología Experimental de Cuyo (IMBECU), Mendoza, Argentina

* ja_mariani@hotmail.com

**Data Availability Statement:** All relevant data are within the manuscript and it's Supporting Information files.

## Abstract

### Background

The role of oral vitamin $D_3$ supplementation for hospitalized patients with COVID-19 remains to be determined. The study was aimed to evaluate whether vitamin $D_3$ supplementation could prevent respiratory worsening among hospitalized patients with COVID-19.

### Methods and findings

We designed a multicentre, randomized, double-blind, sequential, placebo-controlled clinical trial. The study was conducted in 17 second and third level hospitals, located in four provinces of Argentina, from 14 August 2020 to 22 June 2021. We enrolled 218 adult

**Funding:** This study was supported by the National Agency for the Promotion of Research, Technological Development and Innovation (grant FONCyT IP COVID-19-931). Vitamin D3 and placebo were donated by Raffo S.A., an argentinian pharmaceutical company. The funders had no role in the design and conduct of the study; collection, management, analysis, and interpretation of the data; preparation, review, or approval of the manuscript; and decision to submit the manuscript for publication.

**Competing interests:** The authors have declared that no competing interests exist. Raffo S.A. provided support in the form of vitamin D3 and placebo capsules. This does not alter our adherence to PLOS ONE policies on sharing data and materials. There are no patents, products in development or marketed products associated with this research to declare.

patients, hospitalized in general wards with SARS-CoV-2 confirmed infection, mild-to-moderate COVID-19 and risk factors for disease progression. Participants were randomized to a single oral dose of 500 000 IU of vitamin $D_3$ or matching placebo. Randomization ratio was 1:1, with permuted blocks and stratified for study site, diabetes and age ($\leq$60 vs >60 years). The primary outcome was the change in the respiratory Sepsis related Organ Failure Assessment score between baseline and the highest value recorded up to day 7. Secondary outcomes included the length of hospital stay; intensive care unit admission; and in-hospital mortality. Overall, 115 participants were assigned to vitamin $D_3$ and 105 to placebo (mean [SD] age, 59.1 [10.7] years; 103 [47.2%] women). There were no significant differences in the primary outcome between groups (median [IQR] 0.0 [0.0–1.0] vs 0.0 [0.0–1.0], for vitamin $D_3$ and placebo, respectively; $p$ = 0.925). Median [IQR] length of hospital stay was not significantly different between vitamin $D_3$ group (6.0 [4.0–9.0] days) and placebo group (6.0 [4.0–10.0] days; $p$ = 0.632). There were no significant differences for intensive care unit admissions (7.8% vs 10.7%; RR 0.73; 95% CI 0.32 to 1.70; $p$ = 0.622), or in-hospital mortality (4.3% vs 1.9%; RR 2.24; 95% CI 0.44 to 11.29; $p$ = 0.451). There were no significant differences in serious adverse events (vitamin $D_3$ = 14.8%, placebo = 11.7%).

## Conclusions

Among hospitalized patients with mild-to-moderate COVID-19 and risk factors, a single high oral dose of vitamin $D_3$ as compared with placebo, did not prevent the respiratory worsening.

## Trial registration

ClincicalTrials.gov Identifier: NCT04411446.

## Introduction

Vitamin $D_3$ supplementation has been proposed as a potential therapeutic strategy among patients with coronavirus disease 2019 (COVID-19) [1–3]. Effects of vitamin D that could favourably affect the outcomes of patients with infectious diseases include immunomodulatory and anti-inflammatory actions [4–6]. Acute respiratory disease syndrome is the main cause of death among hospitalized patients with COVID-19, and pro-inflammatory cytokines play a central pathogenic role [7, 8]. Vitamin D reduces pro-inflammatory cytokines, increases those with anti-inflammatory actions, and also upregulates angiotensin-converting enzyme 2 receptor, which is the surface receptor for the entry of severe acute respiratory syndrome coronavirus 2 (SARS-CoV-2) to the alveolar epithelial cells [1, 4–6, 9, 10]. These actions could potentially improve clinical outcomes of patients with COVID-19 pneumonia.

Furthermore, epidemiological studies have suggested a relationship between low vitamin D levels and COVID-19 risk and adverse COVID-19 outcomes, and an open-label pilot clinical trial suggested that supplementation with calcidiol significantly reduced the need for intensive care unit admissions [11–15]. However, a randomized clinical trial involving hospitalized patients with moderate to severe COVID-19 showed no differences in length of hospital stay between oral supplementation with vitamin $D_3$ and placebo [16]. Therefore, the evidence supporting the role of vitamin D supplementation to treat patients with COVID-19 remains inconclusive, particularly among patients with mild to moderate COVID-19 [17–20].

The objective of this randomized, double-blind, placebo-controlled clinical trial was to evaluate whether vitamin $D_3$ supplementation, given as a single high dose, could prevent respiratory worsening among hospitalized patients with mild-to-moderate COVID-19 and risk factors for disease progression.

## Materials and methods

### Study design

The CholecAlcifeRol to improvE the outcomes of patients with COVID-19 (CARED) trial was a multicentre, randomized, double-blind, sequential, placebo-controlled trial, designed by independent investigators and supported by the National Agency for the Promotion of Research, Technological Development and Innovation. The study was conducted in 17 hospitals located in four provinces of Argentina, and the local ethics committees of the participating institutions approved the protocol. Written informed consent was obtained from all participants. The study was conducted in compliance with local regulations on research on human subjects, the Declaration of Helsinki and Good Clinical Practice guidelines. The study is registered in ClinicalTrials.gov (Identifier number NCT04411446). Study protocol has been published elsewhere and is available in S1 File [21]. An independent Data and Safety Monitoring Board (DSMB) monitored the trial. The Consolidated Standards for Reporting Trials (CONSORT) statement was followed to report the study results [22].

The sequential design consisted in an adaptative design with two stages. In the first stage, the study aimed to assess the effects of vitamin D on respiratory Sepsis related Organ Failure Assessment (rSOFA), and the second stage aimed to evaluate the effects of vitamin D on clinical events. The protocol specified that the decision to proceed to the second stage would be made conditioned by the results of the primary endpoint analysis of the first stage.

### Participants

The participants were adults aged 18 or older patients and either gender, who had been admitted to general wards in the last 24 hours, with SARS-CoV-2 confirmed infection by reverse transcriptase–polymerase chain reaction, an expected hospitalization for at least 24 hours, oxygen saturation ≥90% (measured by pulse oximetry) breathing ambient air, and at least one of the following conditions: age 45 or older or hypertension, diabetes, chronic obstructive pulmonary disease or asthma (at least moderate), cardiovascular disease (history of myocardial infarction, percutaneous transluminal coronary angioplasty, coronary artery bypass grafting or valve replacement surgery) or body mass index ≥30 (S1 File). Age 45 or older was selected as inclusion criterion to ensure a baseline risk of respiratory worsening that allow to detect a therapeutic effect of the treatment, and to preserve the power of the study. Obesity was added as risk condition on October 13, 2020, since it was recognised as risk factor after the study begun. Main exclusion criteria were ≥72 hours since admission, women in childbearing age, requirement for >5 litres/minute of oxygen or mechanical ventilation, chronic kidney disease requiring haemodialysis or chronic liver failure, chronic supplementation with pharmacological vitamin D, treatment with anticonvulsants, sarcoidosis, malabsorption syndrome, known hypercalcemia, life expectancy <6 months, allergy to study medication, or any condition at discretion of investigator impeding to understand the study and give informed consent (S1 File).

### Randomization and intervention

After giving informed consent, patients were randomly assigned in a 1:1 ratio to receive vitamin $D_3$ (cholecalciferol) or matching placebo, using an interactive web response system with

permuted blocks of size 16 and 24. Randomization was stratified by study site, diabetes (yes vs no) and age ($\leq$60 vs >60 years). Castor®, and electronic data capture plataform that has online randomization capability, was used for randomization and data collection (https://www.castoredc.com).

Study interventions consisted in a single oral dose of 500 000 IU of vitamin $D_3$ soft gel capsules (5 capsules of 100 000 IU) or matching placebo, given as soon as possible after randomization.

The study medication was packaged, labelled and shipped to the research sites by pharmacists from the Faculty of Pharmacy of Maimonides University. The pharmacy staff had no other role in the study, nor did they have contact with the sites.

## Outcomes and follow-up

Follow-up was limited to hospitalization. During the first seven days blood pressure, heart rate, pulse oximetry (SpO2), temperature, inspired fraction of oxygen (FiO2), respiratory rate, and clinical and adverse events were recorded. In the cases that remained hospitalized for more than seven days, clinical and adverse events were recorded from day 8 until day 30, the discharge or death, whichever occurred first.

The primary outcome was the change in the rSOFA between baseline and the highest rSOFA recorded up to day 7. The rSOFA was calculated by using the SpO2 instead the partial pressure of oxygen in arterial blood (PaO2), since was expected that most patients would not have arterial blood draws during hospitalization [23–25]. The rSOFA score was calculated with participant breathing room air, however, for participants with oxygen supplementation requirement and for whom treating physician judged inappropriate to temporary interrupt, a guide for FiO2 estimation was provided to investigators (S2 File). Values of ratios SpO2/FiO2 for rSOFA calculations were as follows: > = 400, rSOFA 0; <400 and > = 300, rSOFA 1; <300 and > = 200, rSOFA 2; <200 and > = 100, rSOFA 3; <100, rSOFA 4.

Secondary outcomes included the change in SpO2 between baseline and the lower value recorded during the first seven days; desaturation, defined as SpO2 $\leq$ 90%; the combined endpoint of oxygen supplementation >40%, non-invasive mechanical ventilation or invasive mechanical ventilation (this was the primary outcome of the second stage in the case the study proceed); the change in the quick SOFA between baseline and the highest value recorded during the first 7 days [26]; the requirement of invasive mechanical ventilation; the intensive care unit admission (ICU); the length of hospital stay; the ICU length of stay; acute kidney injury; and the in-hospital mortality.

Serious adverse events were defined as any occurrence in a participant that caused death, was life-threatening, prolonged the hospitalization, caused significant or persistent disability and/or was judged by investigators to represent a significant risk for participant.

A sample of 16 participants from two study sites had blood samples draws for measurement of serum 25-hydroxyvitamin D (25-OH VitD), at baseline and after 3 to 7 days after treatment. Serum 25-OH VitD levels were determined quantitatively by chemiluminescence immunoassay in a central laboratory (A98856, Access 25(OH) Vitamin D Total, Beckman Coulter Inc., USA) [27].

## Statistical analysis

For the first stage, it was estimated that 168 patients would give the trial 80% power to detect a between study groups difference of one point in the change of rSOFA, assuming a standard deviation (SD) of 2, and a type I error of 5%. The sample size was increased to 200 patients to account for non-adherence with the protocol.

Analyses were conducted according to the intention to treat principle.

Continuous data are expressed as means and SD in cases where normal distribution held, and medians and interquartile ranges otherwise. Categorical data are presented as frequencies and percentages. To compare continuous variables, the Student's T-test or the Mann-Whitney U test, as appropriate, was used. Normality assumption was assessed using histograms and Shapiro-Wilk's test. Categorical variables were compared using Pearson's Chi2 test or Exact Fisher's test, as appropriate. Continuous outcomes are presented as differences in medians with the corresponding 95% confidence intervals (95% CI). Differences in medians and the confidence intervals for these differences were generated using smoothed bootstrap with 5000 replicates. Categorical outcomes are presented as risk ratios and 95% CIs. For primary outcome, the Wilcoxon-Mann-Whitney odds (WMWOdds) with the corresponding 95% CIs was computed [28].

Pre-specified subgroups included age ($\geq$60 vs <60 years), gender, diabetes, hypertension, cardiovascular disease, body mass index (>30 vs $\leq$30) and smoking status (current vs former or never). Subgroup analyses were carried out using ordinal regression models with an interaction term of the subgroup indicator variable by treatment.

For primary outcome, a sensitivity analysis using ordinal regression models was carried out, adjusting the estimated treatment effects for stratification variables (site, diabetes and age). Also, a post-hoc adjusted analysis using a ordinal regression model to account for imbalances in COPD and asthma distribution was carried out.

All tests are 2-sided and a *p* value <0.05 was considered as statistically significant.

Analyses were conducted using R version 4.1.0 (R: A language and environment for statistical computing. R Foundation for Statistical Computing, Vienna, Austria).

## Termination of the trial

As prespecified in the protocol, a blinded analysis was carried out after the recruitment of the first 200 participants, the estimated sample size for the initial stage of the study. After a revision of the results, the Steering Committee decided to stop the recruitment and terminate the trial on 7th July 2021. This decision was based on that the differences between groups, either on the primary outcome (i.e., the change in rSOFA) and the secondary outcomes, did not meet the prespecified criteria to proceed to the second stage (the minimum difference considered in the protocol was 0.3 points in rSOFA -S1 File). The decision was communicated thereafter to the DSMB and to the local investigators.

## Ethics

The Ethics Committee of the Hospital El Cruce (Comité de Ética en Investigación Hospital de Alta Complejidad El Cruce) approved the study on 23 June 2020 (reference 36/2020). The local ethics committees of the participating institutions approved the study protocol before the start of the trial in each sites.

## Results

Two hundred eighteen participants were included in the study between August 2020 and June 2021, at 17 research sites located in four provinces of Argentina (S2 File). One hundred fifteen and one hundred three patients were randomly assigned to vitamin $D_3$ and to placebo, respectively. One patient with SpO2<90% at admission was erroneously recruited and randomly assigned to vitamin $D_3$ group, this participant was included in final analysis (Fig 1).

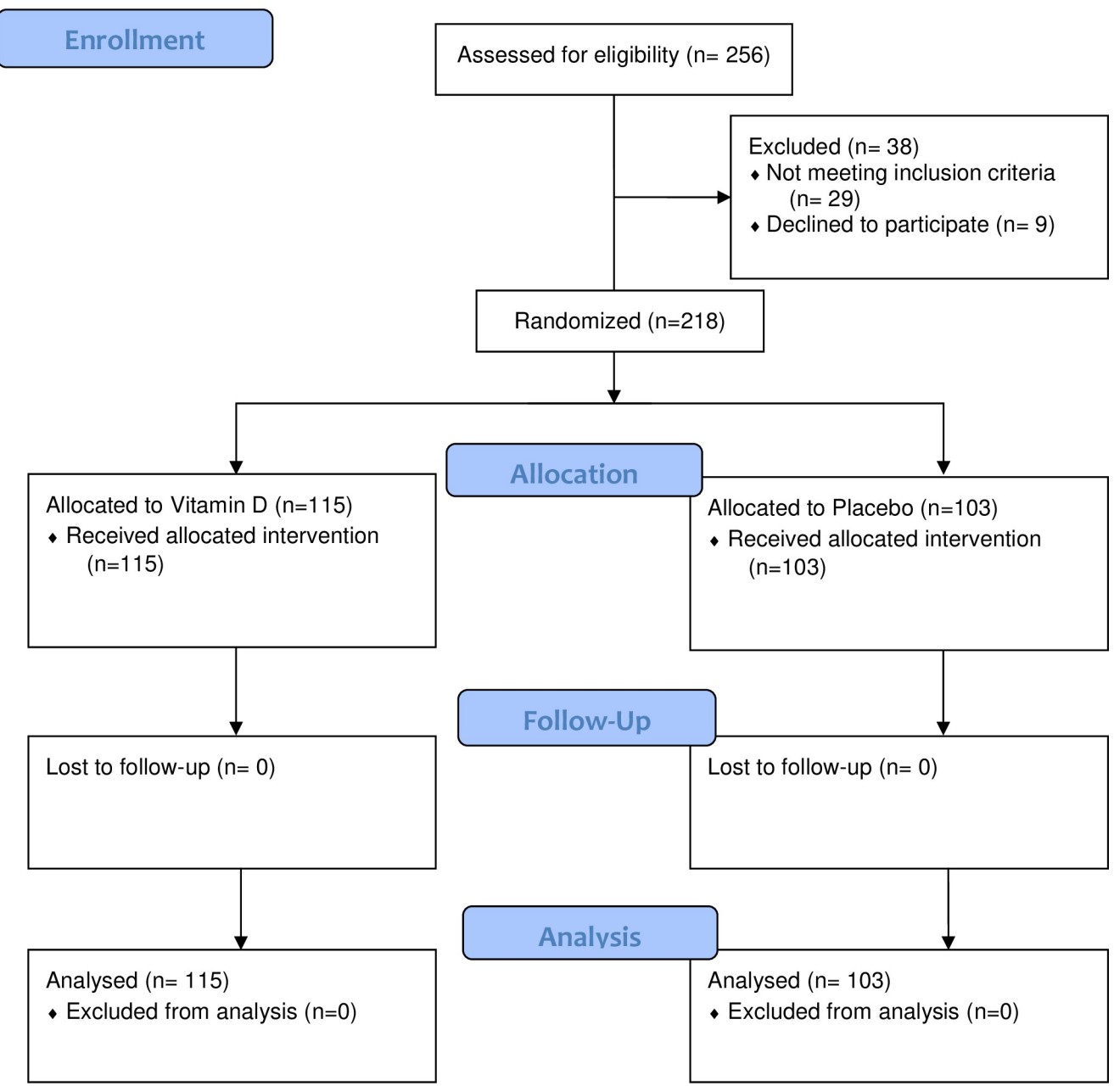

**Fig 1. Flow of the patients in the CARED study.**

### Baseline characteristics

Mean age was 59.1 (10.6) years and 103 (47.2%) patients were women. Risk factors included diabetes in 58 (26.6%) patients, hypertension in 94 (43.1%) patients, obesity in 87 (39.9%), chronic respiratory disease in 26 (11.9%) patients, and cardiovascular disease in 10 (4.6%). Median time from onset of the symptoms to admission was 7.0 (IQR 5.0 to 10.0) days, 194 (89.0%) patients had pneumonia, mean SpO2 was 95.3% (2.0%), 47 (21.6%) patients required oxygen supplementation at enrolment (Table 1). Median time from hospital admission to randomization was 1.0 (IQR 1.0 to 2.0) days. There were no significant differences between treatment groups in baseline characteristics (Table 1).

**Table 1. Participants characteristics.**

| Variables | Vitamin D$_3$ | Placebo |
|---|---|---|
| No. (%) with data | 115 (100) | 103 (100) |
| Time from admission to randomization[a], days | 1.0 (1.0–2.0) | 2.0 (1.0–2.0) |
| Age, mean (SD), y | 59.8 (10.7) | 58.3 (10.6) |
| Women, No. (%) | 51 (44.3) | 52 (50.5) |
| Body mass index[a] | 28.4 (25.8–32.8) | 27.7 (25.6–31.6) |
| Hypertension, No. (%) | 47 (40.9) | 47 (45.6) |
| Diabetes, No. (%) | 32 (27.8) | 26 (25.2) |
| Smoking, No. (%) | | |
| Never | 80 (69.6) | 74 (71.8) |
| Former | 30 (26.1) | 26 (25.2) |
| Current | 5 (4.3) | 3 (2.9) |
| Asthma or Chronic obstructive pulmonary disease, No. (%) | 17 (14.8) | 9 (8.7) |
| Cardiovascular disease, No. (%) | 6 (5.2) | 4 (3.9) |
| Hypothyroidism, No. (%) | 14 (12.2) | 11 (10.7) |
| Neoplasm, No. (%) | 0 (0.0) | 2 (1.9) |
| COVID-19 symptoms | | |
| Dyspnea, No. (%) | 55 (47.8) | 45 (43.7) |
| Fever, No. (%) | 80 (69.6) | 68 (66.0) |
| Symptoms onset to admission[a], days | 7.0 (5.0–10.0) | 8.0 (5.5–10.0) |
| Anosmia, No. (%) | 38 (33.0) | 34 (33.0) |
| Pneumonia, No. (%) | 105 (91.3) | 89 (86.4) |
| Diarrhea, No. (%) | 29 (25.2) | 23 (22.3) |
| Myalgia, No. (%) | 65 (56.5) | 42 (40.8) |
| Physical examination | | |
| Heart rate[a], beats/min | 78.0 (72.0–90.0) | 79.0 (70.5–90.0) |
| Respiratory rate[a], breaths/min | 18.0 (18.0–20.0) | 18.0 (18.0–20.0) |
| Pulse oximetry[a], % | 95.0 (94.0–97.0) | 96.0 (94.0–97.0) |
| Oxygen supplementation, No. (%) | 27 (23.5) | 20 (19.4) |
| Laboratory values | | |
| White cell count[a], /mm$^3$ | 5725 (4775–7522) | 5950 (4800–7900) |
| Calcium[a], mg/dL | 8.8 (8.5–9.0) | 8.7 (8.5–8.9) |
| Creatinine clearance[a], mg/ml/1.73 m$^2$ | 86.1 (73.2–102.4) | 85.6 (70.6–111.1) |
| 25-hydroxyvitamin Vitamin D[ab], ng/mL | 32.5 (27.2–44.2) | 30.5 (22.5–36.2) |

Abbreviations: IQR, interquartile range, COVID-19, coronavirus disease 2019.

SI conversion factors: to convert calcium to mmol/L, multiply by 0.25; 25-hydroxyvitamin vitamin D to nmol/L, multiply by 2.496.

[a]Median (IQR).

[b]16 participants with data.

## Vitamin D supplementation

Study medications were adequately tolerated and there were no reports of immediate adverse reactions after study capsules intake. A subset of 16 patients had 25-OH VitD levels measured at baseline and after study treatment (median 6 [IQR 3.4 to 6.0] days). Baseline measurements, were 32.5 ng/ml (IQR 27.2 to 44.2) and 30.5 ng/ml (IQR 22.5 to 36.2), for the vitamin D$_3$ and placebo group, respectively ($p$ = 0.416). Post-treatment 25-OH VitD levels were 102.0 ng/ml (IQR 85.2 to 132.2) and 30.0 ng/ml (IQR 27.5 to 31.0), for vitamin D$_3$ and placebo group, respectively ($p$ = 0.001).

## Primary outcome

A change in the rSOFA up to day 7 was recorded in 65 participants (29.8%), in 64 patients increased in at least one point, and in one patient decreased one point. There were no significant differences between groups in the distribution of changes in rSOFA (Fig 2A). Among participants in vitamin D3 group, 0.9%, 70.4%, 13.0%, 4.3%, 8.7% and 2.6% had a change in rSOFA of -1, 0, 1, 2, 3 and 4 points, respectively; theses percentages for participants in the placebo group were 0.0%, 69.9%, 15.5%, 5.8%, 4.9%, 3.9%. The median (IQR) of the change in the rSOFA was 0.0 (0.0 to 1.0) points in the vitamin $D_3$ group and 0.0 (0.0 to 0.0) in the placebo group (between-group difference 0.00; 95% CI -0.18 to 0.15; $p$ = 0.825) (WMWOdds 0.97; 95% CI 0.76–1.24) (Table 2).

## Secondary outcomes

The median (IQR) of the change in SpO2 between baseline and the lowest value recorded during the first 7 days was -1.0% (-3.0 to 0.0) and -1% (-4.0 to 0.0) among participants in the vitamin $D_3$ group and placebo group, respectively ($p$ = 0.952) (Fig 2B). Median length of hospital stay was 6.0 (IQR 4.0 to 9.0) days and 6.0 (IQR 4.0 to 10.0) days in the vitamin $D_3$ and placebo groups, respectively ($p$ = 0.614).

Overall, 20 (9.2%) patients were admitted to the ICU and 7 (3.2%) died, without significant differences between groups (Table 2). There were no significant differences between groups in other secondary outcomes (Table 2).

## Subgroups

Results for the primary outcome were consistent across all prespecified subgroups. There were no significant interactions between subgroup indicators and treatment effects (Fig 3). These analyses should be carefully interpreted since the number of participants in each subgroup is small.

## Sensitivity analysis

The analysis adjusting the estimated effects of treatment for stratification variables gave similar results to main analysis (OR 0.96; 95% CI 0.70 to 1.31; p = 0.805).

The results of a *post-hoc* analysis adjusting for the imbalance in the distribution of COPD or asthma between groups, were similar to main results (OR 0.99; 95% CI 0.69 to 1.41; p = 0.950).

## Safety

Overall, 45 serious adverse events among 29 participants were reported. There were no significant differences between groups in either the occurrence of at least one serious adverse event or in the incidence of specific events by organ (Table 3).

# Discussion

The results of the present study show that a single, high, oral dose of vitamin $D_3$ among a non-selected population of patients with mild-to-moderate COVID-19 did not prevent the respiratory worsening. Also, there were no significant effects on the length of hospital stay or other outcomes. Furthermore, overall results were consistent across all pre-specified subgroups.

Previous studies have suggested a role of vitamin D supplementation for the prevention of acute respiratory disease, particularly among individuals with low serum 25-OH VitD levels [29–31]. Several mechanisms have been proposed for the potential beneficial effects of vitamin

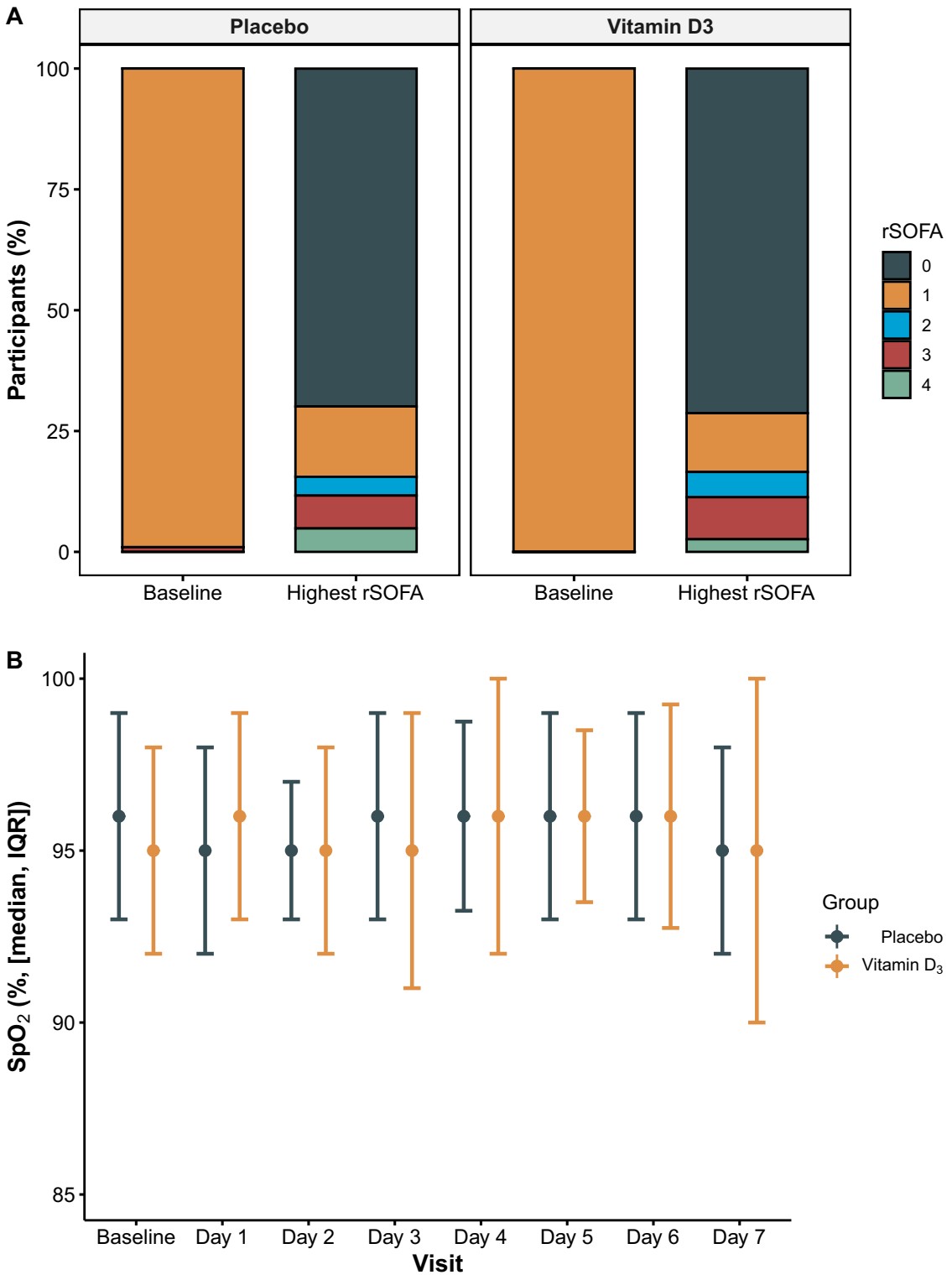

**Fig 2.** Changes in rSOFA scale from baseline to wort value recorded (A), and SpO2 during first week (B).

**Table 2. Study outcomes.**

| Outcomes | Vitamin D₃ (n = 115) | Placebo (n = 103) | Between-group difference (95% CI)[ac] | P |
|---|---|---|---|---|
| | Median (IQR) | Median (IQR) | | |
| Change in rSOFA from baseline[b] | 0.0 (0.0–1.0) | 0.0 (0.0–1.0) | 0.00 (-0.18 to 0.15) | 0.825 |
| Change in SpO₂, % | -1.0 (-3.0–0.0) | -1.0 (-4.0–0.0) | 0.00 (-0.87 to 1.02) | 0.952 |
| Change in quick SOFA | 0.0 (0.0–1.0) | 0.0 (0.0–1.0) | 0.00 (-0.25 to 0.30) | 0.990 |
| Length of stay, days | 6.0 (4.0–9.0) | 6.0 (4.0–10.0) | 0.00 (-1.84 to 0.95) | 0.632 |
| ICU length of stay, days[c] | 9.0 (5.0–11.1) | 9.0 (4.0–10.8) | 0.00 (-8.31 to 9.71) | 0.909 |
| | No. with events (%) | No. with events (%) | **Risk ratio (95% CI)** | |
| Desaturation | 22 (19.1) | 14 (13.6) | 1.40 (0.76 to 2.60) | 0.359 |
| Oxygen >40%, NIV or MV | 17 (14.8) | 15 (14.6) | 1.02 (0.53 to 1.93) | 1.00 |
| Mechanical ventilation | 5 (4.3) | 6 (5.8) | 0.75 (0.23 to 2.37) | 0.851 |
| Acute kidney injury | 2 (1.7) | 2 (1.9) | 0.90 (0.12 to 6.24) | 1.00 |
| Myocardial infarction | 0 (0.0) | 0 (0.0) | - | - |
| Stroke | 0 (0.0) | 0 (0.0) | - | - |
| Pulmonary Embolism | 0 (0.0) | 0 (0.) | - | - |
| ICU admission | 9 (7.8) | 11 (10.7) | 0.73 (0.32 to 1.70) | 0.622 |
| In-hospital Death | 5 (4.3) | 2 (1.9) | 2.24 (0.44 to 11.29) | 0.451 |

Abbreviations: rSOFA, Sepsis related Organ Failure Assessment; SpO2, pulse oximetry; ICU, Intensive Care Unit; NIV, non-invasive ventilation; MV, mechanical ventilation.

[a]Between-group differences are differences in medians and 95% CIs.

[b]Primary outcome.

[c]Differences in medians with their corresponding 95% CIs were obtained using smoothed bootstrap.

[d]Among 20 patients that were admitted to ICU.

D among COVID-19 patients, including the modulation of immune response [1, 7, 9, 32]. Furthermore, previous studies have shown that low vitamin D levels are associated with higher SARS-Cov-2 infection risk, severity and mortality [11–13, 15]. Also, an open-label randomized controlled trial has found that the supplementation with oral calcifediol (0.532 mg at admission, 0.266 on day 3 and 7, and 0.266 weekly until discharge) could improve clinical outcomes among hospitalized patients [14]. However, a double-blind trial among moderate to severe COVID-19 patients, showed that a single oral dose supplementation with 200 000 IU of vitamin D₃ did not reduce the length of hospital stay [16]. The present study is in line with these results and extends them to mild-to-moderate COVID-19 patients, and to other relevant outcomes (i.e., the degree of respiratory worsening).

The study used a single dose of 500 000 IU of oral vitamin D₃ since it was previously demonstrated that this scheme rapidly increases plasma levels of 25-OH VitD, and that achieved levels are maintained for at least 4 weeks, covering the period of highest risk for respiratory worsening, with an adequate security profile [33]. Although, an increase in 25-OH VitD plasma levels was achieved, there were no effects of treatment on the study outcomes. These results are consistent with another trial showing that there were no significant effects of vitamin D₃ supplementation despite an increase in plasma 25-OH VitD, neither in all cohort nor in 25-OH VitD deficient participants [16]. It has been hypothesized that a supplementation with a single high dose of vitamin D, although it serves to raise plasma levels of 25-OH VitD, does not improve the immune response, and that chronic supplementation, with daily or weekly doses of vitamin D would produce better clinical results instead [30, 34]. Perhaps, the stage of the disease at hospial admission in the present study was too late for treatment to express benefial effects, since vitamin D needs several days to induce the mechanisms immune,

| Subgroup | OR (95% CI) | | P for interaction |
|---|---|---|---|
| Age | | | 0.922 |
| >60 years | 0.98 ( 0.64 to 1.5 ) | | |
| <=60 years | 0.95 ( 0.65 to 1.4 ) | | |
| Gender | | | 0.995 |
| Males | 0.98 ( 0.66 to 1.46 ) | | |
| Females | 0.97 ( 0.64 to 1.47 ) | | |
| Diabetes | | | 0.430 |
| Yes | 0.79 ( 0.46 to 1.37 ) | | |
| No | 1.05 ( 0.75 to 1.47 ) | | |
| Body Mass Index | | | 0.890 |
| >30 | 1.02 ( 0.65 to 1.61 ) | | |
| <=30 | 0.97 ( 0.67 to 1.4 ) | | |
| Cardiovascular disease | | | 0.408 |
| Yes | 0.41 ( 0.1 to 1.75 ) | | |
| No | 1.01 ( 0.75 to 1.35 ) | | |
| Smoking | | | 0.929 |
| Current | 0.85 ( 0.15 to 4.91 ) | | |
| Former/Never | 0.98 ( 0.73 to 1.31 ) | | |
| Overall | 0.98 ( 0.74 to 1.3 ) | | 0.896 |

0  0.5  1  2        5
OR

**Fig 3. Subgroup analyses.**

**Table 3. Serious adverse events.**

|  | Vitamin D$_3$ | Placebo | P |
|---|---|---|---|
| No. with data | 115 | 103 | |
| At least one seriuos adverse event, No. (%) | 17 (14.8) | 12 (11.7) | 0.631 |
| Cardiovascular, No. (%) | 6 (5.2) | 4 (3.9) | 0.884 |
| Metabolic, No. (%) | 3 (2.6) | 2 (1.9) | 1.00 |
| Infectious, No. (%) | 5 (4.3) | 3 (2.9) | 0.840 |
| Respiratory, No. (%) | 2 (1.7) | 2 (1.9) | 1.00 |
| Hematologic, No. (%) | 2 (1.7) | 1 (1.0) | 1.00 |
| Gastrointestinal, No. (%) | 7 (6.1) | 5 (4.9) | 0.920 |
| Neurological, No. (%) | 3 (1.4) | 0 (0.0) | 0.249 |

metabolics, antioxidant, endocrine leading to antiviral effects [20]. However, well-designed clinical trials are necessary to evaluate the effects of different regimens of vitamin D supplementation.

Additionally, it is worthwhile that the baseline 25-OH VitD plasma levels were higher than in other studies [11, 16, 17]; these differences could be due to differences in populations studied and to the fact that in our study the 25-OH VitD measurements were conducted mainly during the summer and early autumn, but the causes remain speculative [35]. Although results of vitamin D3 supplementation for treatment of patients with COVID-19 could be theoretically modified by the serum vitamin D status, with deficient populations obtaining the most benefits, this remains speculative. Moreover, two clinical trial that included critically ill patients -most of them with infectious diseases- with vitamin D defficiency (≤20 ng/mL) and randomized them to high vitamin D$_3$ doses or placebo showed no beneficial effect of the treatment [36, 37].

The strengths of this study include the double-blind, placebo-controlled design, the multicentre inclusion of participants with representation of a broad sociocultural background from the country, and the high adherence to study protocol and follow-up.

The present study has several limitations. A single high dose of vitamin D$_3$ was chosen to ensure rapid and persistent high plasma levels of 25-OH VitD, it is possible that multiple dosing regimens could have different biological effects [30]. The primary outcome assessed the effects of the treatment on the respiratory system, precluding to detect other potentially relevant effects. The follow-up was limited to hospital stay, longer follow-up would be necessary to detect relevant effects on recovery after discharge. Also, the study was underpowered to detect differences between groups on clinically important events (i.e., intensive care unit admission, mechanical ventilation, mortality). Participants were admitted with a median of 7 days from symptoms onset and most of them with established pneumonia; whether treatment earlier in the course of disease could modify the subsequent clinical course has yet to be determined [38]. In the present study, the measured serum 25-OH VitD levels among the participants with blood samples were sufficient, whether different results would be obtained among a vitamin D deficient population remains to be determined.

The SpO2/FiO2 ratio used as primary outcome have been validated as surrogate of PO2/FiO2. Although validation studies of SpO2/FiO2 ratio did not included patients with COVID-19, the absence of effects on other measures of respiratory worsening besides rSOFA, gives reassurance to study results [24, 25, 39]. Since women of childbearing age were excluded from the study our results are not generalizable to this population.

## Conclusions

Supplementation with a single, high dose of vitamin $D_3$ at admission to patients hospitalized with mild-to-moderate COVID-19 did not prevent respiratory worsening as compared with placebo.

## Supporting information

**S1 Checklist.**
(DOC)

**S1 File. Protocol of the study.**
(PDF)

**S2 File. Guide for FiO2 estimation for participants with oxygen supplementation.**
(PDF)

**S3 File. Sites, principal investigators and number of participants recruited in the study.**
(PDF)

**S4 File. Dataset of the primary analysis.**
(XLSX)

## Author Contributions

**Conceptualization:** Javier Mariani, Laura Antonietti, Carlos Tajer, León Ferder, Felipe Inserra, Walter Manucha.

**Data curation:** Javier Mariani, Laura Antonietti, Milagro Sanchez Cunto, Fernando Ross, Lucila Moracho.

**Formal analysis:** Javier Mariani.

**Funding acquisition:** Javier Mariani, Laura Antonietti, Carlos Tajer, León Ferder, Felipe Inserra, Walter Manucha.

**Investigation:** Javier Mariani, Laura Antonietti, Carlos Tajer, León Ferder, Felipe Inserra, Milagro Sanchez Cunto, Diego Brosio, Fernando Ross, Marcelo Zylberman, Daniel Emilio López, Cecilia Luna Hisano, Sebastián Maristany Batisda, Gabriela Pace, Adrián Salvatore, Jimena Fernanda Hogrefe, Marcela Turela, Andrés Gaido, Beatriz Rodera, Elizabeth Banega, María Eugenia Iglesias, Mariela Rzepeski, Juan Manuel Gomez Portillo, Magalí Bertelli, Andrés Vilela, Leandro Heffner, Verónica Laura Annetta, Lucila Moracho, Maximiliano Carmona, Graciela Melito, María José Martínez, Gloria Luna, Natalia Vensentini, Walter Manucha.

**Methodology:** Javier Mariani, Laura Antonietti, Carlos Tajer, León Ferder, Felipe Inserra, Graciela Melito, Natalia Vensentini, Walter Manucha.

**Project administration:** Javier Mariani, Laura Antonietti, Maximiliano Carmona, Natalia Vensentini, Walter Manucha.

**Resources:** Javier Mariani, Laura Antonietti, León Ferder, Felipe Inserra, Maximiliano Carmona, Graciela Melito, María José Martínez, Natalia Vensentini, Walter Manucha.

**Software:** Javier Mariani.

**Supervision:** Javier Mariani, Laura Antonietti, Carlos Tajer, Maximiliano Carmona, María José Martínez, Gloria Luna, Natalia Vensentini, Walter Manucha.

**Validation:** Javier Mariani, Laura Antonietti, Milagro Sanchez Cunto, Diego Brosio, Fernando Ross, Marcelo Zylberman, María Eugenia Iglesias, Mariela Rzepeski, Juan Manuel Gomez Portillo, Magalí Bertelli, Andrés Vilela, Leandro Heffner, Verónica Laura Annetta, Lucila Moracho, Maximiliano Carmona, Graciela Melito, María José Martínez, Gloria Luna, Natalia Vensentini, Walter Manucha.

**Visualization:** Javier Mariani, Laura Antonietti, Carlos Tajer, León Ferder, Felipe Inserra, Milagro Sanchez Cunto, Diego Brosio, Fernando Ross, Marcelo Zylberman, Daniel Emilio López, Cecilia Luna Hisano, Sebastián Maristany Batisda, Gabriela Pace, Adrián Salvatore, Jimena Fernanda Hogrefe, Marcela Turela, Andrés Gaido, Beatriz Rodera, Elizabeth Banega, María Eugenia Iglesias, Mariela Rzepeski, Juan Manuel Gomez Portillo, Magalí Bertelli, Andrés Vilela, Leandro Heffner, Verónica Laura Annetta, Lucila Moracho, Maximiliano Carmona, Graciela Melito, María José Martínez, Gloria Luna, Natalia Vensentini, Walter Manucha.

**Writing – original draft:** Javier Mariani, Laura Antonietti, Carlos Tajer, León Ferder, Felipe Inserra, Milagro Sanchez Cunto, Diego Brosio, Fernando Ross, Marcelo Zylberman, Daniel Emilio López, Cecilia Luna Hisano, Sebastián Maristany Batisda, Gabriela Pace, Adrián Salvatore, Jimena Fernanda Hogrefe, Marcela Turela, Andrés Gaido, Beatriz Rodera, Elizabeth Banega, María Eugenia Iglesias, Mariela Rzepeski, Juan Manuel Gomez Portillo, Magalí Bertelli, Andrés Vilela, Leandro Heffner, Verónica Laura Annetta, Lucila Moracho, Maximiliano Carmona, Graciela Melito, María José Martínez, Gloria Luna, Natalia Vensentini, Walter Manucha.

**Writing – review & editing:** Javier Mariani, Laura Antonietti, Carlos Tajer, León Ferder, Felipe Inserra, Milagro Sanchez Cunto, Diego Brosio, Fernando Ross, Marcelo Zylberman, Daniel Emilio López, Cecilia Luna Hisano, Sebastián Maristany Batisda, Gabriela Pace, Adrián Salvatore, Jimena Fernanda Hogrefe, Marcela Turela, Andrés Gaido, Beatriz Rodera, Elizabeth Banega, María Eugenia Iglesias, Mariela Rzepeski, Juan Manuel Gomez Portillo, Magalí Bertelli, Andrés Vilela, Leandro Heffner, Verónica Laura Annetta, Lucila Moracho, Maximiliano Carmona, Graciela Melito, María José Martínez, Gloria Luna, Natalia Vensentini, Walter Manucha.

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
