## [Decision Letter · Decision Letter 0]

3 Feb 2022

PONE-D-21-40739High-dose vitamin D versus placebo to prevent complications in COVID-19 patients: multicentre randomized controlled clinical trialPLOS ONE

Dear Dr. Mariani,

Thank you for submitting your manuscript to PLOS ONE. After careful consideration, we feel that it has merit but does not fully meet PLOS ONE’s publication criteria as it currently stands. Therefore, we invite you to submit a revised version of the manuscript that addresses the points raised during the review process.

We look forward to receiving your revised manuscript.

Kind regards,

Alessandro Putzu, M.D.

Academic Editor

PLOS ONE

Journal Requirements:

“This study was supported by the National Agency for the Promotion of Research, Technological Development and Innovation (grant FONCyT IP COVID-19-931). Vitamin D3 and placebo were donated by Raffo S.A., an argentinian pharmaceutical company. The funders had no role in the design and conduct of the study; collection, management, analysis, and interpretation of the data; preparation, review, or approval of the manuscript; and decision to submit the manuscript for publication.”

Please note that funding information should not appear in other areas of your manuscript. We will only publish funding information present in the Funding Statement section of the online submission form.

“This study was supported by the National Agency for the Promotion of Research, Technological Development and Innovation (grant FONCyT IP COVID-19-931). Vitamin D3 and placebo were donated by Raffo S.A., an argentinian pharmaceutical company. The funders had no role in the design and conduct of the study; collection, management, analysis, and interpretation of the data; preparation, review, or approval of the manuscript; and decision to submit the manuscript for publication.”

Additional Editor Comments:

Thank you for your submission to PLOS ONE. A number of issues have been identified in the review process. While we feel that this manuscript shows promise, we also think that a major revision is needed. Before we can make a final decision about this manuscript we want to offer you the opportunity to revise and resubmit the manuscript.

Please clearly report in the manuscript (or supplementary material) any change to methods after trial commencement, with reasons.

Reviewers' comments:

Reviewer's Responses to Questions

**Comments to the Author**

1. Is the manuscript technically sound, and do the data support the conclusions?

Reviewer #1: Yes

Reviewer #2: Partly

Reviewer #3: Yes

Reviewer #4: Partly

Reviewer #5: Yes

Reviewer #6: Yes

2. Has the statistical analysis been performed appropriately and rigorously? 

Reviewer #1: Yes

Reviewer #2: No

Reviewer #3: No

Reviewer #4: Yes

Reviewer #5: Yes

Reviewer #6: Yes

3. Have the authors made all data underlying the findings in their manuscript fully available?

Reviewer #1: Yes

Reviewer #2: No

Reviewer #3: No

Reviewer #4: No

Reviewer #5: Yes

Reviewer #6: Yes

4. Is the manuscript presented in an intelligible fashion and written in standard English?

Reviewer #1: Yes

Reviewer #2: No

Reviewer #3: Yes

Reviewer #4: Yes

Reviewer #5: Yes

Reviewer #6: Yes

5. Review Comments to the Author

Reviewer #1: The authors report results of multicentre RCT, conducted in 17 hospitals and including 218 adult patients randomized to 500 000 IU of vitamin D3 or placebo. The primary endpoint was the change in the respiratory SOFA score (from SpO2) between baseline and the highest value recorded up to day 7. The authors show that among hospitalized patients with mild-to-moderate COVID-19 and risk factors, a single high oral dose of vitamin D3 versus placebo did not prevent respiratory worsening.

An interesting a well-conducted study, with “negative” results explained by an over-optimistic hypothesis that vitamin D could be a cure to the respiratory manifestations of COVID-19 (this needs to be discussed). The authors, as many others seem convinced that Vit D3 would act as a anti-viral drug in patients who have been sick for about a week before receiving the high-dose vitamin supplement. This is forgetting that micronutrients are not drugs and need to be incorporated into metabolism which requires at least 4-5 days to then exert their antioxidant, metabolic, immune, and endocrine functions. It would be miraculous if VitD3 could be a respiratory cure.

The Ref 29 Martineau-2017-meta-analyis [1] indeed indicates that Vit.D supplementation is safe and protects against acute respiratory tract infection overall, but especially in those who were vitamin D deficient. It was not about cure, but about prevention, and there are other meta showing not effect [2].

Study design – 2nd §: please rewrite – “first stage” and “2nd stage” are probably primary and secondary endpoints.

Page 9: please complete: rSOFA … “since it was…”

Although SpO2 has been validated as a surrogate for the Pa/FiO2 ratio in the SOFA, please provide the ranges of SpO2 used to attribute 0 to 4 points of r-score.

Page 9-last §: Please be more specific about numbers when your write “a sample…”: how many?

Statistics Page 10-2nd §: “normal assumption” is probably “normal distribution” – please replace

Stat Page 10: “would give...” do you mean “would complete the primary endpoint” ?

Page 13: please reorganize the first sentence

Discussion.

Page 15: please change “impairment” to “worsening”.

Page 16: top “other trials” (add “s”)

Very correctly the authors discuss the fact (2nd §) that the baseline VitD levels of their patients did not reveal real deficiencies (study conducted during summer-autumn). This should be emphasized in the discussion, and compared to similar results in critically ill.

Non-ICU patients were enrolled but as large VitD3 trials have been conducted in critically ill patients showing negative results this needs to be discussed [3]. Of note in ICU patients only those with real deficiency might have had benefit on mortality (not the aim here of course) [4].

Limitations: suggestion to discuss the fact that although SpO2 has been validated as a surrogate …. It may not be as specific as the P/FO2 ratio. And of course the absence of vit.D deficit

Table 1: are there results of C-reactive protein? – this CRP is important to determine as VitD, as most micronutrients decrease proportionally to the level of inflammation [5] likely to have been present in your patients, further complicating the interpretation of the results.

Please replace the subtitle “current smoker” by “Smoking”, as current smoker appears below

Please move the last line of table about “Time” to the top of the table under the numbers of patients

Delete all “Median IQR” from the table itself and make it a footnote

Please specific the number of patients with blood determination of VitD

Reference 34: Please complete it

Figure 2 A: the changes (or their absence) in rSOFA are not visible as presented – would be better to show individual values with initial and last.

Does this figure indicate that overall, except for one patient none improved their scores over 7 days? Please comment

Last figure 3 – please change “si” to « yes” on diabetes

1. Martineau A. R., Jolliffe D. A., Hooper R. L., Greenberg L., Aloia J. F., Bergman P. et al. Vitamin D supplementation to prevent acute respiratory tract infections: systematic review and meta-analysis of individual participant data. BMJ 2017; 356:i6583.

2. Pham H., Waterhouse M., Baxter C., Duarte Romero B., McLeod D. S. A., Armstrong B. K. et al. The effect of vitamin D supplementation on acute respiratory tract infection in older Australian adults: an analysis of data from the D-Health Trial. Lancet Diabetes Endocrinol 2021; 9:69-81.

3. Amrein K., Parekh D., Westphal S., Preiser J. C., Berghold A., Riedl R. et al. Effect of high-dose vitamin D3 on 28-day mortality in adult critically ill patients with severe vitamin D deficiency: a study protocol of a multicentre, placebo-controlled double-blind phase III RCT (the VITDALIZE study). BMJ Open 2019; 9:e031083.

4. Holick MF. The vitamin D deficiency pandemic: Approaches for diagnosis, treatment and prevention. Rev Endocr Metab Disord 2017; 18:153-65.

5. Duncan A, Talwar D, McMillan DC, Stefanowicz F, O'Reilly DS. Quantitative data on the magnitude of the systemic inflammatory response and its effect on micronutrient status based on plasma measurements. Am J Clin Nutr 2012; 95:64-71.

Reviewer #2: I appreciate the opportunity to review the manuscript reporting the results of a randomised clinical trial investigating the effect of high-dose vitamin D in patients with COVID-19. The investigators should be commended for completing the trial in the difficult circumstances of the COVID-19 pandemic. I have reviewed the manuscript with the trial protocol, which was published in Trials and registered in ClinicalTrials.gov. Please find my comments below.

1. Please provide how the random allocation sequence was generated and embedded in the web system.

2. Age > 45 was a part of inclusion criteria, and randomisation was stratified by age > 60. Were there any rationales for these cut-offs, particularly for age > 45 in the inclusion criteria?

3. Some of the reported secondary outcomes were different from those supposed to be reported in the trial registration. “The combined end-point of FiO2 > 40%, NIV or invasive MV” was neither registered nor in the published protocol (Trials). This should not be reported as secondary outcomes. If the authors think this needs to be informed, it should be reported as post-hoc analysis.

4. The trial registration listed the following outcomes as secondary outcomes. The results should be reported as defined. Myocardial infarction, stroke, acute kidney injury, pulmonary embolism, the combined outcome, and ICU length of stay.

5. It was unclear how rSOFA was scored using SpO2. Also, Figure 2 indicated that some patients scored as “-1”. This should be explained.

6. Subgroup analyses were not defined in trial registration or pre-published protocol. Furthermore, the reported subgroup analyses differed from what was defined in the investigator’s protocol attached as a supplement. The results of subgroup analysis described in the investigator’s protocol should be reported as such, and the other subgroup analyses should be moved out or reported as post-hoc. Also, conducting such many subgroup analyses in only 200 patients would not be appropriate.

7. The criteria to stop the trial at the first stage were not clearly reported. Please provide the details.

8. The distribution of patients with asthma or COPD appeared to be imbalanced between the groups. The authors may want to add comments and additional analysis incorporating the adjustment for the imbalance.

9. Please provide the rationale for the dosing regimen. Why did the trial give only a bolus dose without daily doses, as the authors might be aware?

Reviewer #3: The authors describe the results of a multi-center, double-blinded RCT of Vitamin D3 supplementation as a potential treatment to prevent respiratory complications among hospitalized COVID-19 patients with mild to moderate infection. They find no evidence that a single high oral dose of vitamin D3 prevented respiratory impairment relative to placebo. While the study design largely appears to be sound, and while this study provides useful additional information on the potential role of vitamin D3 supplementation in the treatment of COVID-19 (a topic on which no clear consensus has, as of yet, emerged), I have some concerns about the statistics used, particularly for the primary outcome and for the subgroup analyses:

Major Comments:

* The 95% confidence intervals for the primary outcome (change in rSOFA score) and several other secondary outcomes (such as change in SpO2) are just single points (95% CI: (0,0)) and so are not useful/nonsensical. Although it is not explicitly stated how the bootstrap was conducted and how these bootstrapped 95% CIs were formed, the bootstrap seems to have been a nonparametric bootstrap (sampling with replacement from the observed data, conditional perhaps on randomization arm or other factors) and the CIs appear to be percentile confidence intervals. Based on the distribution of change scores in Figure 3a, from which we can see that almost 75% of the change scores in both arms are exactly zero, the median change score in each arm is almost guaranteed to be exactly zero in every single one of the bootstrap resamples, leading to a difference in median change score of exactly zero as well. To me, this suggests that the bootstrap distribution is not, in this case, a good approximation to the sampling distribution. To address this issue, it might be worth considering a smoothed bootstrap instead. Alternatively, switching the primary analysis to the Hodges-Lehman estimator and corresponding 95% CI might also be appropriate.

* What was the outcome for the subgroup analysis? The "Subgroups" section of the "Results" section suggests that the subgroup analysis looked at the primary outcome (which was change in rSOFA score from baseline), but the analysis itself was carried out using Poisson regression, which is most appropriate for count data and which does not accommodate negative dependent variables (at least one individual in the dataset had a negative change in rSOFA score from baseline). It is also unclear how one would interpret an IRR for a non-count outcome. As such, if the outcome of interest for the subgroup analysis is in fact change in rSOFA from baseline, I would recommend choosing a different analysis model (e.g., a repeated measures ANCOVA using the baseline and follow-up rSOFA scores, a quantile regression approach using the change score, or an ordinal regression model to capture the odds of observing a score change greater than or equal to some value y).

* Were there any issues with missing data or loss to follow-up? If so, how were they addressed?

* The decision to exclude all women of child-bearing age strikes me as a potential limitation/threat to generalizability. Given that other vitamin d3 trials have not made such exclusions (for example, the RCT of vitamin d3 supplementation in patients with moderate to severe COVID-19 only excluded individuals who were currently pregnant or lactating), why was this decision made?

Minor Comments:

* Please carefully review and revise for grammar. For example, the first line of the abstract reads "The role of oral vitamin D3 supplementation for hospitalized patients with COVID-19 patients remain..." The second "patients" is redundant, and "remain" should be "remains".

Reviewer #4: The authors presented the results of the first stage of a multicentre randomized controlled clinical trial evaluating high-dose vitamin D versus placebo to prevent complications in COVID-19 patients.

The manuscript is well structured and follows the pre-registered and published protocol. I have only some remarks.

The authors stated in the manuscript, that “178 patients would give the first stage of the trial 80% power to detect a between study groups difference of one point in the change of rSOFA, assuming a standard deviation (SD) of 2, and a type I error of 5%.”. Whereas the protocol states that “168 patients would be needed, 84 per group”. I would suggest to clarify the difference in the estimated number of patients. Moreover. I would like to suggest that the assumptions of the sample size estimation be presented. Since a Mann-Whitney U test was planned and used for the analysis of the change in rSOFA score, there should be assumptions on provision probability between groups besides power and significance level.

In addition, it should be noted, that a Mann-Whitney U test does not compare medians (see e.g. Divine et al. DOI: 10.1080/00031305.2017.1305291). Therefore, I would suggest to adjust the presentation of results (median (IQR) of change in rSOFA per group) accordingly.

The assignment of patients to treatment was performed by means of block randomization stratified by site, age and diabetes. It is well known from the literature that stratified randomization leads to correlation between groups. If the analysis does not account for the stratification factors, a loss in power could occur so that the resulting p-values may be too large and confidence intervals to wide. The impact of the adjusted analysis depends on the association between outcome and stratification factor. It may be assumed that the factors are chosen because they are related to the outcome. Therefore, I would recommend to include all stratification factors (site, age and diabetes) in a sensitivity analysis, which accompanies the planned main analysis.

Reviewer #5: This well-written manuscript reports findings of a multi-centre, double-blind, placebo-controlled trial investigating effects of high-dose vitamin D (single dose of 500,000 IU) on clinical outcomes in hospitalised patients with COVID-19 descrbed as ‘moderate to mild’ (though there were 7 deaths out of ~200 participants). The results extend those of Murai et al (JAMA 2020) who showed no effect of 200,000 in moderate to severe COVID-19. Strengths include multi-centre nature of the study, with utilisation of the SOFA score as primary outcome – the European Medicines Agency has accepted that a change in the SOFA score is an acceptable surrogate marker of efficacy in exploratory trials of novel therapeutic agents in sepsis. The study forms an important addition to the literature. There are some limitations, none of which I see as being major:

1. Dosing regimen: the investigators used a very large bolus dose of vitamin D. The advantage is that 25(OH)D levels were elevated quickly. The potential disadvantages are a) safety (25[OH]D levels attained in intervention arm may be associated with toxicity) and b) the unphysiologic rapid elevation then fall in 25(OH)D levels, which may not support immune function optimally. But this is covered in the discussion.

2. Baseline 25D levels were only measured in a subset of participants, which precludes sub-group analysis to see whether effects were seen in those with low baseline vitamin D status. Of note, baseline 25OHD levels were quite high: 32.5 ng/ml (IQR 27.2 to 44.2) and 30.5 ng/ml – so results cant be generalised to populations where deficiency is common. This limitation should be mentioned in the discussion.

3. Inclusion criteria: some are quite subjective, e.g.

‘an expected hospitalization for at least 24 hours’

‘life expectancy <6 months’

‘any condition at discretion of investigator impeding to understand the study’

A bit more detail is needed re how an objective call was made on these.

4. ‘Post-treatment 25-OH VitD levels were 102.0 ng/ml’ – please specify timepoint

5. Were statistical analyses adjusted for stratification factors? They should be.

6. The study had relatively short follow-up – was there any difference in incidence of chronic symptoms / ‘long covid’ between arms?

Reviewer #6: The authors conducted a RCT to evaluate whether vitamin D3 supplementation could prevent respiratory failure in hospitalized patients with mild/moderate COVID-19.

The therapy was not associated with any benefit on the respiratory function.

The trial is well conducted, and the results well presented and well discussed.

6. PLOS authors have the option to publish the peer review history of their article (what does this mean?). If published, this will include your full peer review and any attached files.

Reviewer #1: No

Reviewer #2: No

Reviewer #3: No

Reviewer #4: No

Reviewer #5: No

Reviewer #6: **Yes: **Jean-Louis Vincent

---

## [Author Response · Author response to Decision Letter 0]

4 Mar 2022

Journal Requirements:

Response: Done.

“This study was supported by the National Agency for the Promotion of Research, Technological Development and Innovation (grant FONCyT IP COVID-19-931). Vitamin D3 and placebo were donated by Raffo S.A., an argentinian pharmaceutical company. The funders had no role in the design and conduct of the study; collection, management, analysis, and interpretation of the data; preparation, review, or approval of the manuscript; and decision to submit the manuscript for publication.”

Please note that funding information should not appear in other areas of your manuscript. We will only publish funding information present in the Funding Statement section of the online submission form.

“This study was supported by the National Agency for the Promotion of Research, Technological Development and Innovation (grant FONCyT IP COVID-19-931). Vitamin D3 and placebo were donated by Raffo S.A., an argentinian pharmaceutical company. The funders had no role in the design and conduct of the study; collection, management, analysis, and interpretation of the data; preparation, review, or approval of the manuscript; and decision to submit the manuscript for publication.”

Response: Done.

Response: We changed to "All relevant data are within the manuscript and it's Supporting Information files.". The S3 File contains the treatment and primary outcome variables.

Response: Done (we deleted the statement that was after “Conclusions”).

Response: Done. 

Additional Academic Editor Comments:

Thank you for your submission to PLOS ONE. A number of issues have been identified in the review process. While we feel that this manuscript shows promise, we also think that a major revision is needed. Before we can make a final decision about this manuscript we want to offer you the opportunity to revise and resubmit the manuscript.

Please clearly report in the manuscript (or supplementary material) any change to methods after trial commencement, with reasons.

Response: We added the following sentence in the “Participants” section: “Obesity was added as risk condition on October 13, 2020, since was recognised as risk factor after the study begun.”

 

Reviewer #1

Comment #1: The authors report results of multicentre RCT, conducted in 17 hospitals and including 218 adult patients randomized to 500 000 IU of vitamin D3 or placebo. The primary endpoint was the change in the respiratory SOFA score (from SpO2) between baseline and the highest value recorded up to day 7. The authors show that among hospitalized patients with mild-to-moderate COVID-19 and risk factors, a single high oral dose of vitamin D3 versus placebo did not prevent respiratory worsening.

An interesting a well-conducted study, with “negative” results explained by an over-optimistic hypothesis that vitamin D could be a cure to the respiratory manifestations of COVID-19 (this needs to be discussed). The authors, as many others seem convinced that Vit D3 would act as a anti-viral drug in patients who have been sick for about a week before receiving the high-dose vitamin supplement. This is forgetting that micronutrients are not drugs and need to be incorporated into metabolism which requires at least 4-5 days to then exert their antioxidant, metabolic, immune, and endocrine functions. It would be miraculous if VitD3 could be a respiratory cure.

The Ref 29 Martineau-2017-meta-analyis [1] indeed indicates that Vit.D supplementation is safe and protects against acute respiratory tract infection overall, but especially in those who were vitamin D deficient. It was not about cure, but about prevention, and there are other meta showing not effect [2].

Response #1: Thank you for the comment. We changed the second paragraph of the discussion for “Previous studies have suggested a role of vitamin D supplementation for the prevention of acute respiratory disease, particularly among individuals with low serum 25-OH VitD levels (28, 29, 30)”. We added the reference #30 suggested by the reviewer. Also, we added the sentence in the Discussion “Perhaps, the stage of the disease at hospial admission in the present study was too late for treatment to express benefial effects, since vitamin D needs several days to induce the mechanisms immune, metabolics, antioxidant, endocrine leading to antiviral effects (20).”. 

Comment #2: Study design – 2nd §: please rewrite – “first stage” and “2nd stage” are probably primary and secondary endpoints.

Response #2: Thank you for the comment. Please note that the trial was designed as sequential (or group sequential trial) a type of an adaptative design. For the first stage of the trial we have as primary outcome the change in the rSOFA. If, after the first stage have been completed, the interim analysis would showed a beneficial effect, the trial would procceed to the second stage (wich had as primary outcome the need of high dose of oxygen or mechanical ventilation during hospitalization). For clarification we changed the second paraghaph of the “Design section” to: “The sequential design consisted in an adaptative design with two stages.”

Comment #3: Page 9: please complete: rSOFA … “since it was…”

Response #3: Thanks for the observation. Perhabs the text during conversion was deleted. The complete sentence is: “The rSOFA was calculated by using the SpO2 instead the partial pressure of oxygen in arterial blood (PaO2), since was expected that most patients would not have arterial blood draws during hospitalization (23-25).”

Comment #4: Although SpO2 has been validated as a surrogate for the Pa/FiO2 ratio in the SOFA, please provide the ranges of SpO2 used to attribute 0 to 4 points of r-score.

Response #4: Thank you for the comment. We added the sentence “Values of ratios SpO2/FiO2 for rSOFA calculations were as follows: >=400, rSOFA 0; <400 and >=300, rSOFA 1; <300 and >=200, rSOFA 2; <200 and >=100, rSOFA 3; <100, rSOFA 4.”, in the “Outcomes and follow-up” section.

Comment #5: Page 9-last §: Please be more specific about numbers when your write “a sample…”: how many?

Response #5: Thank you for the observation. We added the number. The sentence is now: “A sample of 16 participants from two study sites had…..”

Comment #6: Statistics Page 10-2nd §: “normal assumption” is probably “normal distribution” – please replace

Response #6: Thank you. Done.

Comment #7: Stat Page 10: “would give...” do you mean “would complete the primary endpoint” ?

Response #7: Thank you very much for the comment. We reformulated this sentence to “For the first stage, it was estimated that 178 patients would give the trial 80% power to detect a between study groups difference of one point in the change of rSOFA”.

Comment #8: Page 13: please reorganize the first sentence

Response #8: Thank you for the observation. We changed to “Two hundred eighteen participants were included in the study between August 2020 and June 2021, at 17 research sites located in four provinces of Argentina (S2 File).”

Comment #9: Page 15: please change “impairment” to “worsening”.

Response #9: Thank you for the comment. We changed in “Discussion” and all along the manuscript.

Comment #10: Page 16: top “other trials” (add “s”)

Response #10: Thank you very much. Please, note that we are referring the Murai IH et al trial. We changed to “another”.

Comment #11: Very correctly the authors discuss the fact (2nd §) that the baseline VitD levels of their patients did not reveal real deficiencies (study conducted during summer-autumn). This should be emphasized in the discussion, and compared to similar results in critically ill.

Non-ICU patients were enrolled but as large VitD3 trials have been conducted in critically ill patients showing negative results this needs to be discussed [3]. Of note in ICU patients only those with real deficiency might have had benefit on mortality (not the aim here of course) [4].

Response #11: Thank you for the comment. We added the following sentence in the “Discussion”: “Although results of vitamin D3 supplementation for treatment of patients with COVID-19 could be teoretically modified by the serum vitamin D status, with defficient populations benefing the most, this remains speculative. Moreover, two clinical trial that included critically ill patients -most of them with infectious diseases- with vitamin D defficiency (≤20 ng/mL) and randomized them to high vitamin D3 doses or placebo showed no beneficial effect of the treatment (35, 36).”. Also, two references (#36, #37) of complited RCT on Vitamin D supplementation for the treatment of critically ill patients with vitamin D defficiency were added: 

- Amrein K, Schnedl C, Holl A, Riedl R, Christopher KB, Pachler C, Urbanic Purkart T, Waltensdorfer A, Münch A, Warnkross H, Stojakovic T, Bisping E, Toller W, Smolle KH, Berghold A, Pieber TR, Dobnig H. Effect of high-dose vitamin D3 on hospital length of stay in critically ill patients with vitamin D deficiency: the VITdAL-ICU randomized clinical trial. JAMA. 2014;312(15):1520-30. doi: 10.1001/jama.2014.13204. 

- National Heart, Lung, and Blood Institute PETAL Clinical Trials Network, Ginde AA, Brower RG, Caterino JM, Finck L, Banner-Goodspeed VM, Grissom CK, Hayden D, Hough CL, Hyzy RC, Khan A, Levitt JE, Park PK, Ringwood N, Rivers EP, Self WH, Shapiro NI, Thompson BT, Yealy DM, Talmor D. Early High-Dose Vitamin D3 for Critically Ill, Vitamin D-Deficient Patients. N Engl J Med. 2019;381(26):2529-2540. doi: 10.1056/NEJMoa1911124. 

Comment #12: Limitations: suggestion to discuss the fact that although SpO2 has been validated as a surrogate …. It may not be as specific as the P/FO2 ratio. And of course the absence of vit.D deficit

Response #12: Thank you for the comment. We added the two limitations noted as follows: “In the present study, the measured serum 25-OH VitD levels were sufficient wether different results would be obtained among a vitamin D deficient population remains to be determined. The SpO2/FiO2 ratio used as primary outcome have been validated as surrogate of PO2/FiO2, although validation studies did not included pacients with COVID-19, our results were consistent for several measures of respiratory worsening (24, 25, 38).” Also, we added a reference (#38): 

- Festic E, Bansal V, Kor DJ, Gajic O; US Critical Illness and Injury Trials Group: Lung Injury Prevention Study Investigators (USCIITG–LIPS). SpO2/FiO2 ratio on hospital admission is an indicator of early acute respiratory distress syndrome development among patients at risk. J Intensive Care Med. 2015;30(4):209-16. doi: 10.1177/0885066613516411. 

Comment #13: Table 1: are there results of C-reactive protein? – this CRP is important to determine as VitD, as most micronutrients decrease proportionally to the level of inflammation [5] likely to have been present in your patients, further complicating the interpretation of the results.

Response #13: Thank you for this observation. Unfortunately we do not have the CRP measurements. As we had severe economic constraints to conduct the trial, we decided to limit at minimum study procedures.

Comment #14: Please replace the subtitle “current smoker” by “Smoking”, as current smoker appears below

Response #14: Thank you. Done.

Comment #15: Please move the last line of table about “Time” to the top of the table under the numbers of patients

Response #15: Thank you. Done.

Comment #16: Delete all “Median IQR” from the table itself and make it a footnote

Response #16: Thank you. Done.

Comment #17: Please specific the number of patients with blood determination of VitD

Response #17: Thank you. Done.

Comment #18: Reference 34: Please complete it

Reponse #18: Thank you. Done.

Comment #19: Figure 2 A: the changes (or their absence) in rSOFA are not visible as presented – would be better to show individual values with initial and last.

Does this figure indicate that overall, except for one patient none improved their scores over 7 days? Please comment

Response #19: Thank you for the comment. We modify the figure to better show the change. The figure shows the change between rSOFA at study entry and the worst rSOFA registered up to 7 days.

Comment #20: Last figure 3 – please change “si” to « yes” on diabetes

Response #20: Thank you for the correction. Done.

Age > 45 was a part of inclusion criteria, and randomisation was stratified by age > 60. Were there any rationales for these cut-offs, particularly for age > 45 in the inclusion criteria?

Reviewer #2

Comment #1: Please provide how the random allocation sequence was generated and embedded in the web system.

Response #1: Thank you for the comment. We used a web system that have a randomization function incorporated, with the option of stratified, permuted-blocks randomization (Castor®). The system generates the sequence. We added the sentence “Castor® was used for randomization and data collection.”, in the “Randomization and intervention” section.

Comment #2: Age > 45 was a part of inclusion criteria, and randomisation was stratified by age > 60. Were there any rationales for these cut-offs, particularly for age > 45 in the inclusion criteria?

Response #2: Thank you for the comment. We stratified by age because, at the time the study was designed, >60 years-old identified a group of particular higher risk for severe COVID-19. The age as inclusion criterion was selected to include population with some degree of clinical risk of in-hospital worsening, and with the potential for take benefits from intervention.

Comment #3: Some of the reported secondary outcomes were different from those supposed to be reported in the trial registration. “The combined end-point of FiO2 > 40%, NIV or invasive MV” was neither registered nor in the published protocol (Trials). This should not be reported as secondary outcomes. If the authors think this needs to be informed, it should be reported as post-hoc analysis.

Response #3: Thank you for the comment. Please note that in ClinicalTrial.gov, this combined end-point is “Need of a high dose of oxygen or mechanical ventilation” and in the definition of the end-point mention “The start of oxygen supplementation at FiO2 >40% or the initiation of invasive through orotracheal intubation) or non-invasive ventilation (Continuous positive airway pressure or Bilevel positive airway ventilation)”. We added “(this was the primary outcome of the second stage in the case the study proceed)” after the definition of the outcome in the manuscritpt.

Comment #4: The trial registration listed the following outcomes as secondary outcomes. The results should be reported as defined. Myocardial infarction, stroke, acute kidney injury, pulmonary embolism, the combined outcome, and ICU length of stay.

Response #4: Thank you very much for the comment. We added the ICU length of stay and acute kidney injury in the “Outcomes” section, also we present the results in Table 2. For other events (stroke, myocardial infarction and pulmonary embolism, there were no events reported. In the case of the combined event, all events are given by the AKI component.

Comment #5: It was unclear how rSOFA was scored using SpO2. Also, Figure 2 indicated that some patients scored as “-1”. This should be explained.

Response #5: Thank you for the comment. We added the values of rSOFA in the “Outcomes” section. Also, as asked by another reviewer, we modified the figure 2A for clarification (the original version of Figure 2A showed the change in rSOFA between baseline and highest -i.e. the worst- value recorded, and the -1 value identified a patient that decreased one point).

Comment #6: Subgroup analyses were not defined in trial registration or pre-published protocol. Furthermore, the reported subgroup analyses differed from what was defined in the investigator’s protocol attached as a supplement. The results of subgroup analysis described in the investigator’s protocol should be reported as such, and the other subgroup analyses should be moved out or reported as post-hoc. Also, conducting such many subgroup analyses in only 200 patients would not be appropriate. 

Response #6: Thank you very much for this comment. As correctly the reviewer pointed out, the subgroup analyses are not described ClinicalTrials.gov. However, are prespecified in the protocol (Supplementary Information 1 at https://www.ncbi.nlm.nih.gov/pmc/articles/PMC7848249/ ). The subgroup “COPD or Asthma” was not prespecified as such, although we prespecified two separate subgroups (for both Asthma, and COPD, but we decided to combine them because there were very small number of participants). In the revised version we removed this subgroup as suggested by the reviewer.

We agree with the observation that only 200 patients would be quite small to conduct subgroup analyses, and added the sentence “These analyses should be carefully interpreted since the number of participants in each subgroup is small.”, in the section to avoid misinterpretations. 

Comment #7: The criteria to stop the trial at the first stage were not clearly reported. Please provide the details.

Response #7: Thank you this comment. We added the sentence “(the minimum difference considered in the protocol was 0.3 points in rSOFA -S1 File)” to reflect what is prespecified in the protocol (page 13 of the S1 File): “An interim analysis will be performed after the incorporation of the first 200 patients. This analysis will aim to assess the effects of treatment on the first stage endpoints. The first stage is designed to have a power of 80% to detect a difference of at least one point in the respiratory SOFA. If a difference of this magnitude or greater is detected, proceed to the second stage of the study. In case of differences between 1 and 0.3 points, the Executive Committee of the study will discuss with the Data Safety and Monitoring Committee the strategy to be followed. This discussion will take into account the magnitude of the Cholecalciferol to improve the evolution of patients with COVID-19 (CARED) Version difference, the effects on other endpoints, the level of significance and the eventual power of the study to detect smaller intervention effects than initially planned.”

Comment #8: The distribution of patients with asthma or COPD appeared to be imbalanced between the groups. The authors may want to add comments and additional analysis incorporating the adjustment for the imbalance.

Reponse #8: Thank you for the observation. As pointed out, there is a difference (although not statistically significant, p value 0.244) in the distribution of asthma or COPD between groups:

The model, adjusting for this imbalance is: 

MODEL FIT:

χ²(2) = 4.141, p = 0.126

Pseudo-R² (Cragg-Uhler) = 0.021

Pseudo-R² (McFadden) = 0.008

AIC = 504.345, BIC = 514.484 

Standard errors: MLE

 exp(Est.) 2.5% 97.5% z val. p

----------- ------- ------- -------- -------

(Intercept) 0.542 0.417 0.705 -4.568 0.000

VitaminD 0.989 0.695 1.406 -0.062 0.950

Asthma-COPD 1.643 1.043 2.590 2.139 0.032

We did not included an adjusted model since it was no prespecified in the protocol and to avoid misinterpretations. If the reviewer prefer we could add the adjusted model as Supplemenal File.

Comment #9: Please provide the rationale for the dosing regimen. Why did the trial give only a bolus dose without daily doses, as the authors might be aware?

Response #9: Thank you for the comment. In the “Discussion” we explain the dose regimen (which was extensively discussed in our study group during the design of the study - in the discussion feasibility issues were raised, besides with the efficacy and safety considerations). We modified the sentence to “The study used a single dose of 500 000 IU of oral vitamin D3 since it was previously demonstrated that this scheme rapidly increases plasma levels of 25-OH VitD, and that achieved levels are maintained for at least 4 weeks, covering the period of highest risk for respiratory worsening, with an adequate security profile [32]”. This explanation is also present in the protocol (please see S1 File, pages 2-3 and 14).

 

Reviewer #3

Comment #1: The 95% confidence intervals for the primary outcome (change in rSOFA score) and several other secondary outcomes (such as change in SpO2) are just single points (95% CI: (0,0)) and so are not useful/nonsensical. Although it is not explicitly stated how the bootstrap was conducted and how these bootstrapped 95% CIs were formed, the bootstrap seems to have been a nonparametric bootstrap (sampling with replacement from the observed data, conditional perhaps on randomization arm or other factors) and the CIs appear to be percentile confidence intervals. Based on the distribution of change scores in Figure 3a, from which we can see that almost 75% of the change scores in both arms are exactly zero, the median change score in each arm is almost guaranteed to be exactly zero in every single one of the bootstrap resamples, leading to a difference in median change score of exactly zero as well. To me, this suggests that the bootstrap distribution is not, in this case, a good approximation to the sampling distribution. To address this issue, it might be worth considering a smoothed bootstrap instead. Alternatively, switching the primary analysis to the Hodges-Lehman estimator and corresponding 95% CI might also be appropriate.

Response #1: Thank you very much for this comment. We reanalized the outcomes using smothed boostrap. This was clarified in the “Statistics” section. Table 2 was modified consequentely.

Comment #2: What was the outcome for the subgroup analysis? The "Subgroups" section of the "Results" section suggests that the subgroup analysis looked at the primary outcome (which was change in rSOFA score from baseline), but the analysis itself was carried out using Poisson regression, which is most appropriate for count data and which does not accommodate negative dependent variables (at least one individual in the dataset had a negative change in rSOFA score from baseline). It is also unclear how one would interpret an IRR for a non-count outcome. As such, if the outcome of interest for the subgroup analysis is in fact change in rSOFA from baseline, I would recommend choosing a different analysis model (e.g., a repeated measures ANCOVA using the baseline and follow-up rSOFA scores, a quantile regression approach using the change score, or an ordinal regression model to capture the odds of observing a score change greater than or equal to some value y).

Response #2: Thank very much for the comment. As observed by the reviewer, the outcome for the subgroup analyses was the change in rSOFA from baseline. We re-analised the data using ordinal regression models. We modified in the “Statistics” section and also in the Figure 3.

Comment #3: Were there any issues with missing data or loss to follow-up? If so, how were they addressed?

Response #3: Thank you for the question. We had no missing data (we selected the change in rSOFA and the worst recorded value as outcome and prespecified an expected hospital stay of at least 24 hs in order to minimize missing information). There were no losts in follow-up.

Comment #4: The decision to exclude all women of child-bearing age strikes me as a potential limitation/threat to generalizability. Given that other vitamin d3 trials have not made such exclusions (for example, the RCT of vitamin d3 supplementation in patients with moderate to severe COVID-19 only excluded individuals who were currently pregnant or lactating), why was this decision made?

Response #4: Thank you for the observation. Since we had resource contraints and the inclusion of pregnant/lactanting womed would probably need mandatory lab examinations, we decided to exclude women in child-bearing age from the study. We added “Since women of childbearing age were excluded from the study our results are not generalizable to this population.” as a limitation in the “Discussion”.

Comment #5: Please carefully review and revise for grammar. For example, the first line of the abstract reads "The role of oral vitamin D3 supplementation for hospitalized patients with COVID-19 patients remain..." The second "patients" is redundant, and "remain" should be "remains".

Response #5: Thank you very much for the comment. We revised the grammar. The pointed out error was corrected.

 

Reviewer #4

Comment #1: The authors stated in the manuscript, that “178 patients would give the first stage of the trial 80% power to detect a between study groups difference of one point in the change of rSOFA, assuming a standard deviation (SD) of 2, and a type I error of 5%.”. Whereas the protocol states that “168 patients would be needed, 84 per group”. I would suggest to clarify the difference in the estimated number of patients. Moreover. I would like to suggest that the assumptions of the sample size estimation be presented. Since a Mann-Whitney U test was planned and used for the analysis of the change in rSOFA score, there should be assumptions on provision probability between groups besides power and significance level.

Response #1: Thank you very much for the comment. We used the following specifications in the expected probabilities of rSOFA change from baseline: 

- Placebo: 0.60, 0.27, 0.10, 0.02, 0.01, for changes of 0, 1, 2, 3 and 4 scale points, respectively.

- Vitamin D: 0.80, 0.14, 0.04 , 0.02, 0.0, for changes of 0, 1, 2, 3 and 4 scale points, respectively.

Also, we specified a ratio vitamin D/placebo 1:1. If the reviewer consider approppriate we could include this information in the manuscript. 

We corrected in the manuscript the “178” to “168”.

Comment #2: In addition, it should be noted, that a Mann-Whitney U test does not compare medians (see e.g. Divine et al. DOI: 10.1080/00031305.2017.1305291). Therefore, I would suggest to adjust the presentation of results (median (IQR) of change in rSOFA per group) accordingly.

Response #2: Thank you very much for the comment. In the revised version we present the values distribution of rSOFA at baseline and at follow-up in figure 2A. We added the description of rSOFA scores changes as “Among participants in vitamin D3 group, 0.9%, 70.4%, 13.0%, 4.3%, 8.7% and 2.6% had a change in rSOFA of -1, 0, 1, 2, 3 and 4 points, respectively; theses percentages for participants in the placebo group were 0.0%, 69.9%, 15.5%, 5.8%, 4.9%, 3.9%.”. 

The results for WMW odds are: 0.97 (0.76-1.24). If the reviewer consider appropriate, we could include this results in the paper.

Comment #3: The assignment of patients to treatment was performed by means of block randomization stratified by site, age and diabetes. It is well known from the literature that stratified randomization leads to correlation between groups. If the analysis does not account for the stratification factors, a loss in power could occur so that the resulting p-values may be too large and confidence intervals to wide. The impact of the adjusted analysis depends on the association between outcome and stratification factor. It may be assumed that the factors are chosen because they are related to the outcome. Therefore, I would recommend to include all stratification factors (site, age and diabetes) in a sensitivity analysis, which accompanies the planned main analysis.

Response #3: Thank you for this comment. We added a sentence in “Statistics”: “For primary outcome, a sensitivity analysis using ordinal regression models was carried out, adjusting the estimated treatment effects for stratification variables (site, diabetes and age). “. Also a “Sensitivity analysis” section with the results of ordinal regressional model adjusted the stratification variables was added.

 

Reviewer #6

Comment #1: Dosing regimen: the investigators used a very large bolus dose of vitamin D. The advantage is that 25(OH)D levels were elevated quickly. The potential disadvantages are a) safety (25[OH]D levels attained in intervention arm may be associated with toxicity) and b) the unphysiologic rapid elevation then fall in 25(OH)D levels, which may not support immune function optimally. But this is covered in the discussion.

Response #1: Thank you very much for this comment.

Comment #2: Baseline 25D levels were only measured in a subset of participants, which precludes sub-group analysis to see whether effects were seen in those with low baseline vitamin D status. Of note, baseline 25OHD levels were quite high: 32.5 ng/ml (IQR 27.2 to 44.2) and 30.5 ng/ml – so results cant be generalised to populations where deficiency is common. This limitation should be mentioned in the discussion.

Response #2: Thank you for the comment. We added the sentence “In the present study, the measured serum 25-OH VitD levels were sufficient wether different results would be obtained among a vitamin D deficient population remains to be determined.”

Comment #3: Inclusion criteria: some are quite subjective, e.g.

‘an expected hospitalization for at least 24 hours’

‘life expectancy <6 months’

‘any condition at discretion of investigator impeding to understand the study’

A bit more detail is needed re how an objective call was made on these.

Response #3: Thank you for this comment. As correctly pointed out by the reviewer some inclusion criteria have a subjective component. The first of them (an expected hospitalization for at least 24 hours) was thinked to avoid the inclusion of patients with very short admissions or planned referral with the consequent missing outcome data. The second criterion refers to terminal conditions that could affect the decision to admit ICU patients (an end-point of the study), in a general context of scarce ICU beds. The third criterion was decided thinking in cognitive disorders, but also in illiterate patients in whom investigators would consider that the patients did not clearly understood the nature of the study (in the time the study was designed, patients were isolated in the hospital, visits were not permitted, and some patients had no contact -neither personal nor by phone- with persons out of hospital). 

Comment #4: ‘Post-treatment 25-OH VitD levels were 102.0 ng/ml’ – please specify timepoint. 

Response #4: Thank you for the observation. Please, note that the sentence above in the same section states “A subset of 16 patients had 25-OH VitD levels measured at baseline and after study treatment (median 6 [IQR 3.4 to 6.0] days).”. If the reviewer prefer, we could modify the order of information.

Commnet #5: Were statistical analyses adjusted for stratification factors? They should be.

Response #5: Thank you very much for this comment. We added a sentivity analysis with an adjusted ordinal regression model.

Comment #6: The study had relatively short follow-up – was there any difference in incidence of chronic symptoms / ‘long covid’ between arms?

Response #6: Thank you for the comment. Unfortunately we have no information at the moment about the incidence (and eventually between-group differences in the incidence) of long COVID in our population. We are planning an epidemiological study (which would include the participants of the trial) for long term evaluation after COVID episode.

 

Reviewer #6

Comment #1: The trial is well conducted, and the results well presented and well discussed.

Response #1: Thank you very much for the comments.

---

## [Decision Letter · Decision Letter 1]

28 Mar 2022

PONE-D-21-40739R1High-dose vitamin D versus placebo to prevent complications in COVID-19 patients: multicentre randomized controlled clinical trialPLOS ONE

Dear Dr. Mariani,

Thank you for submitting your manuscript to PLOS ONE. After careful consideration, we feel that it has merit but does not fully meet PLOS ONE’s publication criteria as it currently stands. Therefore, we invite you to submit a revised version of the manuscript that addresses the points raised during the review process.

We look forward to receiving your revised manuscript.

Kind regards,

Alessandro Putzu, M.D.

Academic Editor

PLOS ONE

Journal Requirements:

Additional Editor Comments:

Thank you for your great work on this manuscript. A number of minor issues have been identified in the review process.

I have sone additional questions regarding adverse events.

- Methods. The definition of adverse event and/or serious adverse event is missing. Death is considered an adverse event?

- Results, adverse events. You are reporting in the text 45 adverse events (or serious adverse events? unclear). However, Table 3 reported only 4 events.

- Table 2. This table refer to serious adverse events, correct? Please correct the caption. Furthermore, I suggest to invert the order of the columns (VitD before placebo, as in Table 1 and 2)

Reviewers' comments:

Reviewer's Responses to Questions

**Comments to the Author**

1. If the authors have adequately addressed your comments raised in a previous round of review and you feel that this manuscript is now acceptable for publication, you may indicate that here to bypass the “Comments to the Author” section, enter your conflict of interest statement in the “Confidential to Editor” section, and submit your "Accept" recommendation.

Reviewer #1: (No Response)

Reviewer #2: (No Response)

Reviewer #3: (No Response)

Reviewer #4: All comments have been addressed

Reviewer #6: All comments have been addressed

2. Is the manuscript technically sound, and do the data support the conclusions?

Reviewer #1: Yes

Reviewer #2: Yes

Reviewer #3: Yes

Reviewer #4: (No Response)

Reviewer #6: Yes

3. Has the statistical analysis been performed appropriately and rigorously? 

Reviewer #1: Yes

Reviewer #2: Yes

Reviewer #3: Yes

Reviewer #4: (No Response)

Reviewer #6: Yes

4. Have the authors made all data underlying the findings in their manuscript fully available?

Reviewer #1: Yes

Reviewer #2: Yes

Reviewer #3: Yes

Reviewer #4: (No Response)

Reviewer #6: Yes

5. Is the manuscript presented in an intelligible fashion and written in standard English?

Reviewer #1: Yes

Reviewer #2: Yes

Reviewer #3: Yes

Reviewer #4: (No Response)

Reviewer #6: Yes

6. Review Comments to the Author

Reviewer #1: Globally the authors have addressed the comments and the study is of interest.

but the rSOFA definition remains problematic: either authors have not understood the question … or they do not want to respond.

The request was to please provide the ranges of SpO2 and FiO2 used to attribute 0 to 4 points of r-score. The answer is the classical P/FO2 ratios ?

“Values of ratios SpO2/FiO2 for rSOFA calculations were as follows: >=400, rSOFA 0; <400 and >=300, rSOFA 1; <… » is an inappropriate answer -

Please profile the SpO2 values that were used to attribute rSOFA scores of 0,… 4.

In my ICU: an SpO2 of 98-100% under 2L/min O2 results would be graded as “1”, etc. (please provide the FiO2 from 2-3-4-5 l/mi O2)

Reviewer #2: I would thank the authors for responding to my comments. The responses sound fair, however, leaving some minor concerns as follows.

1. The authors added, “Castor was used for randomization and data collection” to explain how the randomization and treatment allocation was done. However, this is not clear for readers at all. General (international) readers do not know what Castor was.

2. If any, please provide the rationale that the authors added a threshold of age >45 as a risk for disease progressing. If there is not, and the threshold was merely set arbitrarily, that would be fine but should be mentioned in the manuscript as such.

3. The authors replied that there were no stroke, myocardial infarction, and pulmonary embolism events. However, this was not reported in the manuscript. Please be noted that all outcomes planned to be measured in the protocol should be reported even if there were no events. “No events” itself is an important finding.

4. I would suggest the authors provide the result adjusting for an imbalanced variable (asthma or COPD) in the supplementary materials to support the main finding as additional exploratory analysis.

Reviewer #3: The authors have taken strong efforts to address my initial concerns and comments. I still have two remaining questions about the statistical analysis, as well as a number of grammatical/typographical suggestions.

Statistical Analysis:

* For the main analysis of the primary and secondary endpoints (Table 2), please report the sample statistic for the between-group difference in medians, rather than the mean of the smoothed bootstrap distribution; the bootstrapping procedure is necessary only to quantify uncertainty in/estimate a confidence interval for the point estimate from the sample.

* I still have some questions about missing data and the primary outcome definition. Was the rSOFA measured daily for all trial participants on all days from admission through discharge? Or were there some participants for whom one or more daily measurements were missing? I ask because--while the outcome definition (which focuses on the change from baseline to highest *recorded* post-baseline rSOFA score) does somewhat skirt the issue of missing data---if any individuals were missing rSOFA measurements for one or more post-baseline days, the recorded change potentially underestimates the true change. And if missing data patterns systematically differ between the two study arms, there could be bias as a result. If there are missed/missing measurements in the data, please comment on this missing data/the missing data patterns as well as any implications for your analysis.

Grammatical/Typographical Suggestions:

* Page 5 (revised version): the last sentence of the first paragraph ("Vitamin D reduce pro-inflammatory cytokines, increase those with anti-inflammatory actions...") should be split in two ("these actions could potentially improve clinical outcomes of patients with COVID-19 pneumonia" should be its own sentence) and "reduce"/"increase"/"upregulate" should all be singular ("reduces"/"increases", "upregulates").

* Page 5 (revised version): "the evidence supporting the role of vitamin D supplementation to treat patients with COVID-19 remain inconclusive..." should use "remains"

* Page 8 (revised version): the clause "since was recognized as risk factor after the study begun" is missing an object and should likely read "since it was recognized as a risk factor after the study began"

* Page 15 (revised version): "theses percentages for participants in the placebo group were" contains a typo

* Page 20 (revised version): the sentence "Although results of vitamin D3 supplementation for treatment of patients with COVID-19 could be teoretically modified by the serum vitamin D status, with defficient populations" contains two typos

* Page 21 (revised version): "the study was underpower to detect differences between groups on clinically important events" should read "the study was underpowered"

* Page 21 (revised version): the sentence "the measured serum 25-OH VitD levels were sufficient wether different results" contains a typo and is also incomplete

* Page 21 (revised version): the sentence "The SpO2/FiO2 ratio used as primary outcome have been validated as surrogate of PO2/FiO2, although validation studies did not included pacients with COVID-19, our results were consistent for several measures of respiratory worsening" contains a typo ("pacients"), should be broken into two or more sentences, and should be checked for verb tense and subject/verb agreement

Reviewer #4: I thank the authors for considerung my comments. I would appreciate it if the results were supplemented by the WMW odds.

Reviewer #6: the paper has improved - all comments have been addressed

the text is well written

the study brings interesting results.

7. PLOS authors have the option to publish the peer review history of their article (what does this mean?). If published, this will include your full peer review and any attached files.

Reviewer #1: No

Reviewer #2: No

Reviewer #3: No

Reviewer #4: No

Reviewer #6: **Yes: **jean-louis Vincent

---

## [Author Response · Author response to Decision Letter 1]

4 Apr 2022

Journal Requirements:

1- Response: We reviewed the reference list and searched for retracted articles at: The Retraction Watch Database [Internet]. New York: The Center for Scientific Integrity. 2018. ISSN: 2692-465X.. Available from: http://retractiondatabase.org/. (accessed on April 3rd 2022). We found no retracted articles in the reference list.

 

Additional Editor Comments:

Comment #1: Methods. The definition of adverse event and/or serious adverse event is missing. Death is considered an adverse event?

Response #1: Thank you for the comment. We added the sentence “Serious adverse events were defined as any occurrence in a participant that caused death, was life-threatening, prolonged the hospitalization, caused significant or persistent disability and/or was judged by investigators to represent a significant risk for participant.”, in “Methods”. Death was considered and reported as a Clinical Outcome, not as serious adverse event (i.e. a participant that have had a septic shock that have led to death, has the infectious event reported as a serious adverse event and death as a clinical outcome).

Comment #2: Results, adverse events. You are reporting in the text 45 adverse events (or serious adverse events? unclear). However, Table 3 reported only 4 events.

Response #2: Thank you very much for this comment. We added the word “serious” to all reference to adverse events to avoid confusion. Please, note that Table 3 reports both the number (and percentage) of participants with at least one serious adverse event and the number of events across systems (we added the neurological serious adverse events). The sum of serious adverse events across systems is 45. Please, let us know if there is need for further clarification.

Comment #3: Table 2. This table refer to serious adverse events, correct? Please correct the caption. Furthermore, I suggest to invert the order of the columns (VitD before placebo, as in Table 1 and 2).

Response #3: Thank you for the comment. Table 3 refers to serious adverse events. We corrected the caption. Also, we inverted the order of columns.

 

Reviewers' comments:

Reviewer #1

Comment #1: Globally the authors have addressed the comments and the study is of interest.

but the rSOFA definition remains problematic: either authors have not understood the question … or they do not want to respond. The request was to please provide the ranges of SpO2 and FiO2 used to attribute 0 to 4 points of r-score. The answer is the classical P/FO2 ratios ? “Values of ratios SpO2/FiO2 for rSOFA calculations were as follows: >=400, rSOFA 0; <400 and >=300, rSOFA 1; <… » is an inappropriate answer -

Please profile the SpO2 values that were used to attribute rSOFA scores of 0,… 4.

In my ICU: an SpO2 of 98-100% under 2L/min O2 results would be graded as “1”, etc. (please provide the FiO2 from 2-3-4-5 l/mi O2)

Response #1: Thank you for the comment. Definitely, we did not understood the question. For baseline SpO2 was measured breathing room air and the FiO2 was considered 0.21. For follow-up measures among patients with oxygen supplementation the guide provided to investigators was the following table:

Administration device FiO2 (Liters)

Nasal canulae 0.24-0.28 (Max 5 lts)

Venturi Mask 0.24 (3 lts)

 0.35 (8 lts)

 0.40 (10 lts)

 0.50 (12 lts)

Reservoir mask 0.55 (6 lts)

 0.60 (7 lts)

 0.70 (8 lts)

 0.80 (9 lts)

 0.90-0.99 (10-15 lts)

We added the following sentence in “Outcomes and follow-up”: “The rSOFA score was calculated with participant breathing room air, however, for participants with oxygen supplementation requirement and that treating physician judged not posible to temporary suspend, a guide for FiO2 estimation was provided to investigators (S2 File).”. Also, the table is inclued as Supplemental file, now S2 file.

 

Reviewer #2

Comment #1: The authors added, “Castor was used for randomization and data collection” to explain how the randomization and treatment allocation was done. However, this is not clear for readers at all. General (international) readers do not know what Castor was.

Response #1: Thank you for this comment. We modified the sentence, in the revised version is “Castor®, and electronic data capture plataform that has online randomization capability, was used for randomization and data collection (https://www.castoredc.com).”.

 Comment #2: If any, please provide the rationale that the authors added a threshold of age >45 as a risk for disease progressing. If there is not, and the threshold was merely set arbitrarily, that would be fine but should be mentioned in the manuscript as such.

Response #2: Thank for the comment. We intended to include a population at risk of respiratory worsening during the hospitalization for COVID-19 (the study initiated in a moment of the pandemic when patients were hospilized with a broad spectrum of disease severity, including mild disease). We added the sentence “Age 45 or older was selected as inclusion criterion as an intent to ensure a baseline risk of respiratory worsening that allow to detect a therapeutic effect of the treatment, and to preserve the power of the study.”, in the paragraph describing the inclusion criteria.

Comment #3: The authors replied that there were no stroke, myocardial infarction, and pulmonary embolism events. However, this was not reported in the manuscript. Please be noted that all outcomes planned to be measured in the protocol should be reported even if there were no events. “No events” itself is an important finding.

Response #3: Thank you for the comment. We added the information in the Table 2.

Comment #4: I would suggest the authors provide the result adjusting for an imbalanced variable (asthma or COPD) in the supplementary materials to support the main finding as additional exploratory analysis.

Response #4: Thank you for the comment. We added the results of the adjusted analysis in the “Sensitivity analysis” sub-section of “Results”, with the following sentence: “The results of a post-hoc analysis adjusting for the imbalance in the distribution of COPD or asthma between groups, were similar to main results (OR 0.99; 95% CI 0.69 to 1.41; p=0.950).”

 

Reviewer #3

Comment #1: For the main analysis of the primary and secondary endpoints (Table 2), please report the sample statistic for the between-group difference in medians, rather than the mean of the smoothed bootstrap distribution; the bootstrapping procedure is necessary only to quantify uncertainty in/estimate a confidence interval for the point estimate from the sample.

Response #1: Thank you very much for this comment. We modified in the text and in table 2.

Comment #2: I still have some questions about missing data and the primary outcome definition. Was the rSOFA measured daily for all trial participants on all days from admission through discharge? Or were there some participants for whom one or more daily measurements were missing? I ask because--while the outcome definition (which focuses on the change from baseline to highest *recorded* post-baseline rSOFA score) does somewhat skirt the issue of missing data---if any individuals were missing rSOFA measurements for one or more post-baseline days, the recorded change potentially underestimates the true change. And if missing data patterns systematically differ between the two study arms, there could be bias as a result. If there are missed/missing measurements in the data, please comment on this missing data/the missing data patterns as well as any implications for your analysis.

Response #2: We had no missing data for rSOFA, all participants had their SpO2 and FiO2 measured from baseline to hospital discharge, the death or seventh day from study entry, whichever came first. During the course of the study great emphasis and efforts were devoted to maintain the missingness at minimum.

Comment #3: Grammatical/Typographical Suggestions: Page 5 (revised version): the last sentence of the first paragraph ("Vitamin D reduce pro-inflammatory cytokines, increase those with anti-inflammatory actions...") should be split in two ("these actions could potentially improve clinical outcomes of patients with COVID-19 pneumonia" should be its own sentence) and "reduce"/"increase"/"upregulate" should all be singular ("reduces"/"increases", "upregulates").

Response #3: Thank you for this comment. We splited the sentence and corrected the verbs to singular.

Comment #4: Page 5 (revised version): "the evidence supporting the role of vitamin D supplementation to treat patients with COVID-19 remain inconclusive..." should use "remains"

Response #4: Thank you for the comment. We corrected.

Comment #5: Page 8 (revised version): the clause "since was recognized as risk factor after the study begun" is missing an object and should likely read "since it was recognized as a risk factor after the study began".

Response #5: Thank you for the comment. We corrected.

Comment #6: Page 15 (revised version): "theses percentages for participants in the placebo group were" contains a typo

Response #6: Thank you for the comment. We corrected to “these”.

Comment #7: Page 20 (revised version): the sentence "Although results of vitamin D3 supplementation for treatment of patients with COVID-19 could be teoretically modified by the serum vitamin D status, with defficient populations" contains two typos

Response #7: Thank you for this comment. We corrected and re-formulated the sentence to “Although results of vitamin D3 supplementation for treatment of patients with COVID-19 could be theoretically modified by the serum vitamin D status, with deficient populations obtaining the most benefits, this remains speculative”.

Comment #8: Page 21 (revised version): "the study was underpower to detect differences between groups on clinically important events" should read "the study was underpowered"

Response #8: Thank you for the comment. We corrected.

Comment #9: Page 21 (revised version): the sentence "the measured serum 25-OH VitD levels were sufficient wether different results" contains a typo and is also incomplete.

Response #9: Thank for this observation. We corrected the typo and re-formulated the sentence to “In the present study, the measured serum 25-OH VitD levels among the participants with blood samples were sufficient, whether different results would be obtained among a vitamin D deficient population remains to be determined.”

Comment #10: Page 21 (revised version): the sentence "The SpO2/FiO2 ratio used as primary outcome have been validated as surrogate of PO2/FiO2, although validation studies did not included pacients with COVID-19, our results were consistent for several measures of respiratory worsening" contains a typo ("pacients"), should be broken into two or more sentences, and should be checked for verb tense and subject/verb agreement

Response #10: Thank for this observation. We re-formulated the sentence y two sentences and changed the second part as following to clarify the message: “The SpO2/FiO2 ratio used as primary outcome have been validated as surrogate of PO2/FiO2. Although validation studies of SpO2/FiO2 ratio did not included patients with COVID-19, the absence of effects on other measures of respiratory worsening besides rSOFA, gives reassurance to study results [24,25,38].”

 

Reviewer #4

Comment #1: I thank the authors for considerung my comments. I would appreciate it if the results were supplemented by the WMW odds.

Response #1: Thank you for the comment. We added the sentence “For primary outcome, the Wilcoxon-Mann-Whitney odds (WMWOdds) with the corresponding 95% CIs was computed (28).” In the “Statistical analysis” section; and we added the WMWOdds for the primary outcome in “Results”. Also, we added the cite of Divine GW DOI: 10.1080/00031305.2017.1305291 (reference #28).

---

## [Decision Letter · Decision Letter 2]

19 Apr 2022

High-dose vitamin D versus placebo to prevent complications in COVID-19 patients: multicentre randomized controlled clinical trial

PONE-D-21-40739R2

Dear Dr. Mariani,

We’re pleased to inform you that your manuscript has been judged scientifically suitable for publication and will be formally accepted for publication once it meets all outstanding technical requirements.

Kind regards,

Alessandro Putzu, M.D.

Academic Editor

PLOS ONE

Additional Editor Comments:

Thank you for your further work on this manuscript which now makes a fine contribution to consideration of this important topic.

Reviewers' comments:

Reviewer's Responses to Questions

**Comments to the Author**

1. If the authors have adequately addressed your comments raised in a previous round of review and you feel that this manuscript is now acceptable for publication, you may indicate that here to bypass the “Comments to the Author” section, enter your conflict of interest statement in the “Confidential to Editor” section, and submit your "Accept" recommendation.

Reviewer #2: All comments have been addressed

Reviewer #3: All comments have been addressed

Reviewer #4: All comments have been addressed

2. Is the manuscript technically sound, and do the data support the conclusions?

Reviewer #2: Yes

Reviewer #3: Yes

Reviewer #4: (No Response)

3. Has the statistical analysis been performed appropriately and rigorously? 

Reviewer #2: Yes

Reviewer #3: Yes

Reviewer #4: (No Response)

4. Have the authors made all data underlying the findings in their manuscript fully available?

Reviewer #2: Yes

Reviewer #3: Yes

Reviewer #4: (No Response)

5. Is the manuscript presented in an intelligible fashion and written in standard English?

Reviewer #2: Yes

Reviewer #3: Yes

Reviewer #4: (No Response)

6. Review Comments to the Author

Reviewer #2: (No Response)

Reviewer #3: (No Response)

Reviewer #4: (No Response)

7. PLOS authors have the option to publish the peer review history of their article (what does this mean?). If published, this will include your full peer review and any attached files.

Reviewer #2: No

Reviewer #3: No

Reviewer #4: No

---

## [Editor Report · Acceptance letter]

19 May 2022

PONE-D-21-40739R2 

High-dose vitamin D versus placebo to prevent complications in COVID-19 patients: multicentre randomized controlled clinical trial. 

Dear Dr. Mariani:

I'm pleased to inform you that your manuscript has been deemed suitable for publication in PLOS ONE. Congratulations! Your manuscript is now with our production department. 

Kind regards, 

on behalf of

Dr. Alessandro Putzu 

Academic Editor

PLOS ONE